# Interpretable Unsupervised Diversity Denoising and Artefact Removal

**Mangal Prakash**[1,2]    **Mauricio Delbracio**[3]    **Peyman Milanfar**[3]    **Florian Jug**[1,2,4]

[1]Center for Systems Biology Dresden    [2]Max Planck Institute (CBG)
[3]Google Research    [4]Fondazione Human Technopole
`prakash@mpi-cbg.de, {mdelbra, milanfar}@google.com,`
`florian.jug@fht.org`

## Abstract

Image denoising and artefact removal are complex inverse problems admitting multiple valid solutions. Unsupervised diversity restoration, that is, obtaining a diverse set of possible restorations given a corrupted image, is important for ambiguity removal in many applications such as microscopy where paired data for supervised training are often unobtainable. In real world applications, imaging noise and artefacts are typically hard to model, leading to unsatisfactory performance of existing unsupervised approaches. This work presents an interpretable approach for unsupervised and diverse image restoration. To this end, we introduce a capable architecture called Hierarchical DivNoising (HDN) based on hierarchical Variational Autoencoder. We show that HDN learns an interpretable multi-scale representation of artefacts and we leverage this interpretability to remove imaging artefacts commonly occurring in microscopy data. Our method achieves state-of-the-art results on twelve benchmark image denoising datasets while providing access to a whole distribution of sensibly restored solutions. Additionally, we demonstrate on three real microscopy datasets that HDN removes artefacts without supervision, being the first method capable of doing so while generating multiple plausible restorations all consistent with the given corrupted image.

## 1 Introduction

Deep Learning (DL) based methods are currently the state-of-the-art (SOTA) for image denoising and artefact removal with supervised methods typically showing best performance (Zhang et al., 2017; Weigert et al., 2017; Chen et al., 2018; Delbracio et al., 2020). However, in order to be trained, supervised methods need either paired high and low quality images (Zhang et al., 2017; Weigert et al., 2017; Delbracio et al., 2020) or pairs of low quality images (Lehtinen et al., 2018; Buchholz et al., 2019). Both requirements are often not, or at least not efficiently satisfiable, hindering application of these methods to many practical applications, mainly in microscopy and biomedical imaging. Hence, unsupervised methods that can be trained on noisy images alone Krull et al. (2019); Batson & Royer (2019); Xie et al. (2020); Quan et al. (2020) present an attractive alternative. Methods additionally using models of imaging noise have also been proposed (Krull et al., 2020; Laine et al., 2019; Prakash et al., 2019b) and can further boost the denoising performance of unsupervised models.

Still, unsupervised methods suffer from three drawbacks: $(i)$ they fail to account for the inherent uncertainty present in a corrupted image and produce only a single restored solution (*i.e.* point estimation) [1], $(ii)$ they typically show weaker overall performance than their supervised counterparts, and $(iii)$ they are, by design, limited to pixel-wise noise removal and cannot handle structured noises or other image artefacts (spatially correlated noise).

Recently, the first of these drawbacks was addressed by DivNoising (DN) (Prakash et al., 2021) which proposed a convolutional Variational Autoencoder (VAE) architecture for unsupervised denoising and generates diverse denoised solutions, giving users access to samples from a distribution of sensible denoising results. But DN exhibits poor performance on harder (visually more complex and varied) datasets, *e.g.* diverse sets of natural images. Additionally, it does not improve on

---

[1]This is also true for supervised methods.

the performance achieved with other unsupervised denoisers on existing microscopy benchmark datasets (Prakash et al., 2021), where supervised methods, whenever applicable, lead to best results. Last but not least, DN does not address the problem of artefact removal (structured noise removal).

We hypothesize that the performance of methods like DN is limited by the used VAE architecture, which can neither capture longer range dependencies, nor the full structural complexity of many practically relevant datasets. Although more expressive hierarchical VAE (HVAE) architectures have been known for some time in the context of image generation (Sønderby et al., 2016; Maaløe et al., 2019; Vahdat & Kautz, 2020; Child, 2021), so far they have never been applied to image restoration or denoising. Besides, well performing HVAE architectures are computationally very expensive, with training times easily spanning many days or even weeks on multiple modern GPUs (Vahdat & Kautz, 2020; Child, 2021). Hence, it remains unexplored if more expressive HVAE architectures can indeed improve unsupervised image restoration tasks and if they can do so efficiently enough to justify their application in practical use-cases.

***Contributions.*** In this paper, we first introduce a new architecture for unsupervised diversity denoising called HIERARCHICAL DIVNOISING (HDN), inspired by ladder VAEs (Sønderby et al., 2016), but introducing a number of important task-specific modifications. We show that HDN is considerably more expressive than DN (see Fig. 2), leading to much improved denoising results (see Fig. 1), while still not being excessively computationally expensive (on a common 6 GB Tesla P100 GPU training still only takes about 1 day). Most importantly though, HDN leads to SOTA results on natural images and biomedical image data, outperforming 8 popular unsupervised denoising baselines and considerably closing the gap to supervised results (see Table 1).

Additionally, we investigate the application of diversity denoising to the removal of spatially correlated structured noise in microscopy images, an application of immense practical impact. More specifically, we show that HDN, owing to its hierarchical architecture, enforces an interpretable decomposition of learned representations and we devise an effective approach to remove structured noise in microscopy data by taking full advantage of this interpretable hierarchical representation. We showcase this on multiple real-world datasets from diverse microscopy modalities, demonstrating that our method is an important step towards interpretable image restoration and is immediately applicable to many real-world datasets as they are acquired in the life-sciences on a daily basis.

***Related Work.*** Apart from the DL based works already discussed earlier, classical methods such as Non-Local Means (Buades et al., 2005) and BM3D (Dabov et al., 2007) have also found widespread use for denoising applications. A detailed review of classical denoising methods can be found in Milanfar (2012). An interesting unsupervised DL based image restoration method is Deep Image Prior (DIP) (Ulyanov et al., 2018) which showed that convolutional neural networks (CNNs) can be used to restore corrupted images without supervision when training is stopped at an apriori unknown time before convergence. Methods such as DIP (or other methods that require training for each input image separately, *e.g.* SELF2SELF (Quan et al., 2020)) are, however, computationally demanding when applied to entire datasets. Diversity pixel denoising based on VAEs was introduced by DN (Prakash et al., 2021), and a similar approach using GANs by Ohayon et al. (2021).

For structured noise removal, Broaddus et al. (2020) extended Krull et al. (2019) to specifically remove structured line artefacts that can arise in microscopy data. Their method needs exact apriori knowledge about the size of artefacts, which cannot span more than some connected pixels. Dorta et al. (2018) model structured noises as the uncertainty in the VAE decoder represented by a structured Gaussian distribution with non-diagonal covariance matrix which is learned separately by another network. Jancsary et al. (2012) learn a structured noise model in a regression tree field framework. Unlike these approaches, we take an interpretability-first perspective for artefact removal and remove artefacts even without having/learning any structured noise model.

## 2 THE IMAGE RESTORATION TASK

The task of image restoration involves the estimation of a clean and unobservable signal $\mathbf{s} = (s_1, s_2, \ldots, s_N)$ from a corrupted reconstruction $\mathbf{x} = (x_1, x_2, \ldots, x_N)$, where $s_i$ and $x_i$, refer to the respective pixel intensities in the image domain. In general, the reconstructed image comes from solving an inverse imaging problem giving by the forward model,

$$\mathbf{y} = A(\mathbf{s}) + \mathbf{e}, \tag{1}$$

where $A$ is the forward operator (tomography, blurring, sub-sampling, etc.), $\mathbf{e}$ is noise in the measurements typically assumed iid, and $\mathbf{y}$ is the noisy measurements. An image reconstruction algorithm is needed to recover an estimation $\mathbf{x}$ of $\mathbf{s}$ from the noisy measurements $\mathbf{y}$.

Typically, the image reconstruction is obtained through an optimization formulation, where we seek a solution $\mathbf{x}$ that fits the observed values and is compatible with some image prior $R$,

$$\mathbf{x} = \arg\min_{\mathbf{s}'} \|A(\mathbf{s}') - \mathbf{y}\|^2 + \lambda R(\mathbf{s}'), \tag{2}$$

where $\mathbf{s}'$ is the auxiliary variable for the signal being optimized for and $\lambda \geqslant 0$ is related to the level of confidence of the prior $R$. There exists an extensive amount of work defining image priors (Rudin et al., 1992; Golub et al., 1999; Bruckstein et al., 2009; Zoran & Weiss, 2011; Romano et al., 2017; Ulyanov et al., 2018).

Without loss of generality, we can decompose the reconstructed image as $\mathbf{x} = \mathbf{s} + \mathbf{n}$, where $\mathbf{n}$ is the residual (noise) between the ideal image and the reconstructed one. Generally, the *noise* $\mathbf{n}$ on the reconstruction $\mathbf{x}$ is composed of pixel-noise (such as Poisson or Gaussian noise) and multi-pixel artefacts or structured noise that affect groups of pixels in correlated ways. Such artefacts arise through dependencies that are introduced by the adopted reconstruction technique and the domain-specific inverse problem (*e.g.* tomography, microscopy, or ISP in an optical camera). For example, consider the case where the reconstruction is done using Tikhonov regularization $R(\mathbf{s}) = \|\mathbf{s}\|^2$, in the particular case of linear operator. In this case, the solution to Eq. 2 is $\mathbf{x} = (A^T A + \lambda I)^{-1} A^T y$. Thus,

$$\mathbf{x} = (A^T A + \lambda I)^{-1} A^T A \mathbf{s} + (A^T A + \lambda I)^{-1} A^T e = \mathbf{s} + \mathbf{n}, \tag{3}$$

where

$$\mathbf{n} = ((A^T A + \lambda I)^{-1} A^T A - I)\mathbf{s} + (A^T A + \lambda I)^{-1} A^T \mathbf{e}. \tag{4}$$

The reconstructed image is affected by colored noise (term in Eq. 4 depending on $\mathbf{e}$), and also by a structured perturbation introduced by the reconstruction method (term in Eq. 4 depending on $\mathbf{s}$). This structured perturbation appears even in the case when the measurements are noiseless. Accurately modeling the noise/artefacts on the reconstructed image is therefore challenging even in the simplest case where the reconstruction is linear, showing that structured noise removal is non-trivial.

In Section 5, we deal with image denoising where noise contribution $n_i$ to each pixel $x_i$ is assumed to be conditionally independent given the signal $s_i$, *i.e.* $p(\mathbf{n}|\mathbf{s}) = \prod_i p(n_i|s_i)$ (Krull et al., 2019). In many practical applications, including the ones presented in Section 6, this assumption does not hold true, and the noise $\mathbf{n}$ is referred to as structured noise/artefacts.

## 3    NON-HIERARCHICAL VAE BASED DIVERSITY DENOISERS

Recently, non-hierarchical VAE based architectures (Kingma & Welling, 2014; Rezende et al., 2014; Prakash et al., 2021) were explored in the context of image denoising allowing a key advantage of providing samples $s^k$ drawn from a posterior distribution of denoised images. More formally, the denoising operation of these models $f_\psi$ can be expressed by $f_\psi(\mathbf{x}) = s^k \sim p(\mathbf{s}|\mathbf{x})$, where $\psi$ are the model parameters. In this section, we provide a brief overview of these models.

*Vanilla VAEs.* VAEs are generative encoder-decoder models, capable of learning a latent representation of the data and capturing a distribution over inputs $\mathbf{x}$ (Kingma & Welling, 2019; Rezende et al., 2014). The encoder maps input $\mathbf{x}$ to a conditional distribution $q_\phi(\mathbf{z}|\mathbf{x})$ in latent space. The decoder, $g_\theta(\mathbf{z})$, takes a sample from $q_\phi(\mathbf{z}|\mathbf{x})$ and maps it to a distribution $p_\theta(\mathbf{x}|\mathbf{z})$ in image space. Encoder and decoder are neural networks, jointly trained to minimize the loss

$$\mathcal{L}_{\phi,\theta}(\mathbf{x}) = \mathbb{E}_{q_\phi(\mathbf{z}|\mathbf{x})}[-\log p_\theta(\mathbf{x}|\mathbf{z})] + \mathbb{KL}\left(q_\phi(\mathbf{z}|\mathbf{x})\|p(\mathbf{z})\right) = \mathcal{L}_{\phi,\theta}^{\text{R}}(\mathbf{x}) + \mathcal{L}_\phi^{\text{KL}}(\mathbf{x}), \tag{5}$$

with the second term being the KL-divergence between the encoder distribution $q_\phi(\mathbf{z}|\mathbf{x})$ and prior distribution $p(\mathbf{z})$ (usually a unit Normal distribution). The network parameters of the encoder and decoder are given by $\phi$ and $\theta$, respectively. The decoder usually factorizes over pixels as

$$p_\theta(\mathbf{x}|\mathbf{z}) = \prod_{i=1}^{N} p_\theta(x_i|\mathbf{z}), \tag{6}$$

with $p_\theta(x_i|\mathbf{z})$ being a Normal distribution predicted by the decoder, $x_i$ representing the intensity of pixel $i$, and $N$ being the total number of pixels in image $\mathbf{x}$. The encoder distribution is modeled in a similar way, factorizing over the dimensions of the latent space $\mathbf{z}$. Performance of vanilla VAEs on pixel-noise removal tasks is analyzed and compared to DN in Prakash et al. (2021).

**DivNoising (DN).** DN (Prakash et al., 2021) is a VAE based method for unsupervised pixel noise removal that incorporates an explicit pixel wise noise model $p_{NM}(\mathbf{x}|\mathbf{s})$ in the decoder. More formally, the generic Normal distribution over pixel intensities of Eq. 6 is replaced with a known noise model $p_{NM}(\mathbf{x}|\mathbf{s})$ which factorizes as a product of pixels, *i.e.* $p_{NM}(\mathbf{x}|\mathbf{s}) = \prod_i^N p_{NM}(x_i|s_i)$, and the decoder learns a mapping from $\mathbf{z}$ directly to the space of restored images, *i.e.* $g_\theta(\mathbf{z}) = \mathbf{s}$. Therefore,

$$p_\theta(\mathbf{x}|\mathbf{z}) = p_{NM}(\mathbf{x}|g_\theta(\mathbf{z})). \tag{7}$$

The loss of DN hence becomes

$$\mathcal{L}_{\phi,\theta}(\mathbf{x}) = \mathbb{E}_{q_\phi(\mathbf{z}|\mathbf{x})}\left[\sum_{i=1}^N -\log p(x_i|g_\theta(\mathbf{z}))\right] + \mathbb{KL}\left(q_\phi(\mathbf{z}|\mathbf{x})||p(\mathbf{z})\right) = \mathcal{L}_{\phi,\theta}^R(\mathbf{x}) + \mathcal{L}_\phi^{KL}(\mathbf{x}). \tag{8}$$

Intuitively, the reconstruction loss in Eq. 8 measures how likely the generated image is given the noise model, while the KL loss incentivizes the encoded distribution to be unit Normal. DN is the current SOTA for many unsupervised denoising benchmarks (particularly working well for microscopy images), but performs poorly for complex domains such as natural image denoising (Prakash et al., 2021).

## 4 Hierarchical DivNoising

Here we introduce a new diversity denoising setup called Hierarchical DivNoising (HDN) built on the ideas of hierarchical VAEs (Sønderby et al., 2016; Maaløe et al., 2019; Vahdat & Kautz, 2020; Child, 2021). HDN splits its latent representation over $n > 1$ hierarchically dependent stochastic latent variables $\mathbf{z} = \{\mathbf{z}_1, \mathbf{z}_2, ..., \mathbf{z}_n\}$. As in (Child, 2021), $\mathbf{z}_1$ is the bottom-most latent variable (see Appendix Fig. 5), seeing the input essentially in unaltered form. The top-most latent variable is $\mathbf{z}_n$, receiving an $n - 1$ times downsampled input and starting a cascade of latent variable conditioning back down the hierarchy. The architecture consists of two paths: a *bottom-up* path and a *top-down* path implemented by neural networks with parameters $\phi$ and $\theta$, respectively. The *bottom-up* path extracts features from a given input at different scales, while the *top-down* path processes latent variables at different scales top-to-bottom conditioned on the latent representation of the layers above.

The *top-down* network performs a downward pass to compute a hierarchical prior $p_\theta(\mathbf{z})$ which factorizes as

$$p_\theta(\mathbf{z}) = p_\theta(\mathbf{z}_n) \prod_{i=1}^{n-1} p_\theta(\mathbf{z}_i|\mathbf{z}_{j>i}), \tag{9}$$

with each factor $p_\theta(\mathbf{z}_i|\mathbf{z}_{j>i})$ being a learned multivariate Normal distribution with diagonal covariance, and $\mathbf{z}_{j>i}$ referring to all $\mathbf{z}_j$ with $j > i$.

Given a noisy input $\mathbf{x}$, the encoder distribution $q_\phi(\mathbf{z}|\mathbf{x})$ is computed in two steps. First the *bottom-up* network performs a deterministic upward pass and extracts representations from noisy input $\mathbf{x}$ at each layer. Next, the representations extracted by the *top-down* network during the downward pass are merged with results from the *bottom-up* network before inferring the conditional distribution

$$q_\phi(\mathbf{z}|\mathbf{x}) = q_\phi(\mathbf{z}_n|\mathbf{x}) \prod_i^{n-1} q_{\phi,\theta}(\mathbf{z}_i|\mathbf{z}_{j>i}, \mathbf{x}). \tag{10}$$

The conditional distribution $p_\theta(\mathbf{x}|\mathbf{z})$ is described in Eq. 7. The generative model along with the prior $p_\theta(\mathbf{z})$ and noise model $p_{NM}(\mathbf{x}|\mathbf{s})$ describes the joint distribution

$$p_\theta(\mathbf{z}, \mathbf{x}, \mathbf{s}) = p_{NM}(\mathbf{x}|\mathbf{s})p_\theta(\mathbf{s}|\mathbf{z})p_\theta(\mathbf{z}). \tag{11}$$

Considering Eq. 7 and 9, the denoising loss function for HDN thus becomes

$$\mathcal{L}_{\phi,\theta}(\mathbf{x}) = \mathcal{L}_{\phi,\theta}^R(\mathbf{x}) + \mathbb{KL}\left(q_\phi(\mathbf{z}_n|\mathbf{x})||p_\theta(\mathbf{z}_n)\right) + \sum_{i=1}^{n-1} \mathbb{E}_{q_\phi(\mathbf{z}_{j>i}|\mathbf{x})}[\mathbb{KL}\left(q_{\phi,\theta}(\mathbf{z}_i|\mathbf{z}_{j>i}, \mathbf{x})||p_\theta(\mathbf{z}_i|\mathbf{z}_{j>i}))\right], \tag{12}$$

with $\mathcal{L}_{\phi,\theta}^R(\mathbf{x})$ being the same as in Eq. 8. Similar to DN, the reconstruction loss in Eq. 12 measures how likely the generated image is under the given noise model, while the KL loss is calculated between the conditional prior and approximate posterior at each layer.

Following Vahdat & Kautz (2020); Child (2021), we use gated residual blocks in the encoder and decoder. Additionally, as proposed in Maaløe et al. (2019); Liévin et al. (2019), our generative path uses skip connections which enforce conditioning on all layers above. We also found that

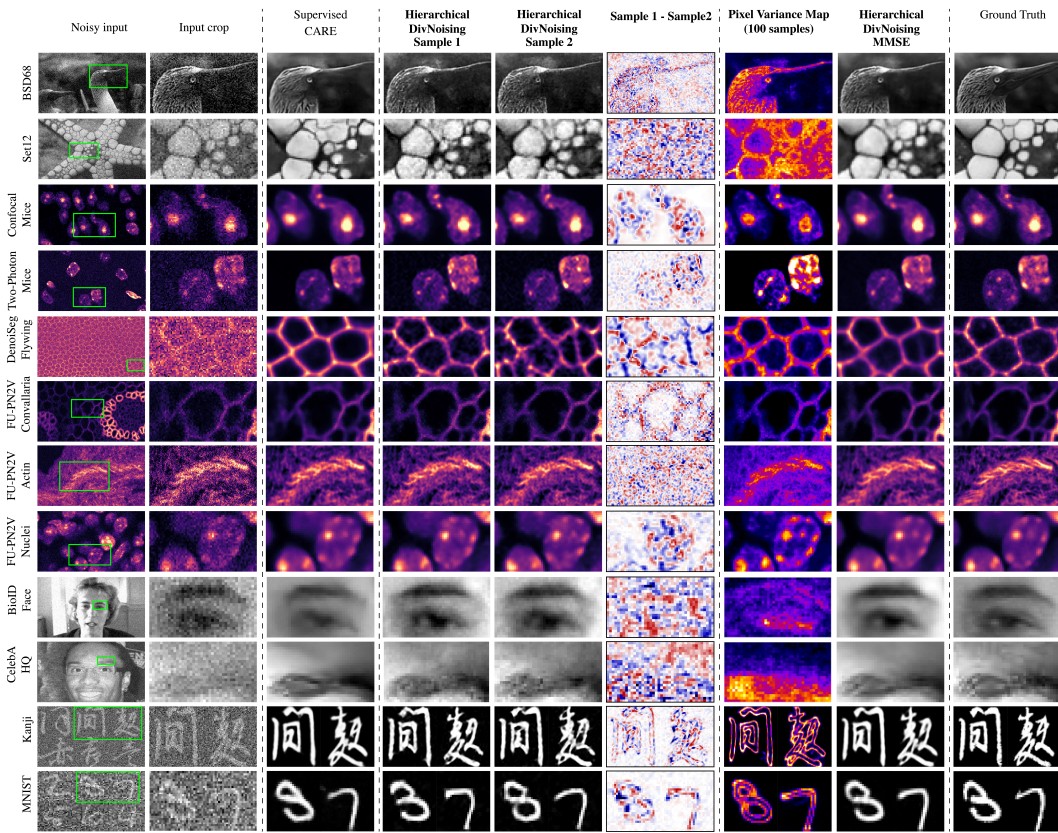

Figure 1: **Qualitative pixel-noise removal with HIERARCHICAL DIVNOISING (HDN).** For a subset of the datasets we tested *(rows)*, we show denoising results for one representative input image. Columns show, from left to right, the input image and an interesting crop region, results of CARE, two randomly chosen HDN samples, a difference map between those samples, the per-pixel variance of 100 diverse samples, the MMSE estimate derived by averaging these 100 samples, and the ground truth image.

employing batch normalization (Ioffe & Szegedy, 2015) additionally boosts performance. To avoid *KL-vanishing* (Bowman et al., 2015), we use the so called *free-bits* approach (Kingma et al., 2016; Chen et al., 2016), which defines a threshold on the KL term in Eq. 12 and then only uses this term if its value lies above the threshold. A schematic of our fully convolutional network architecture is shown in Appendix A.1.

***Noise Model.*** Our noise model factorizes as $p_{\mathrm{NM}}(\mathbf{x}|\mathbf{s}) = \prod_i^N p_{\mathrm{NM}}(x_i|s_i)$, implying that the corruption occurs independently per pixel given the signal, which is only true for pixel-noises (such as Gaussian and Poisson). Pixel noise model can easily be obtained from either calibration images (Krull et al., 2020; Prakash et al., 2019b) or via bootstrapping from noisy data (Prakash et al., 2019b; 2021). Interestingly, despite using such pixel noise models, we will later see that HDN can also be used to remove structured noises (see Section 6).

## 5    DENOISING WITH HDN

To evaluate our unsupervised diversity denoising method, we used 12 publicly available denoising datasets from different image domains, including natural images, microscopy data, face images, and hand-written characters. Some of these datasets are synthetically corrupted with pixel-noise while others are intrinsically noisy. Examples of noisy images for each dataset are shown in Fig. 1. Appendix A.1.1 describes all datasets in detail.

We compare HDN on all datasets against $(i)$ 7 unsupervised baseline methods, *i.e.* BM3D (Dabov et al., 2007), Deep Image Prior (DIP) (Ulyanov et al., 2018), SELF2SELF (S2S) (Quan et al., 2020), NOISE2VOID (N2V) (Krull et al., 2019), Probabilistic NOISE2VOID (PN2V) (Krull et al., 2020), NOISE2SAME (Xie et al., 2020), and DIVNOISING (DN) (Prakash et al., 2021); and $(ii)$ against 2 supervised methods, namely NOISE2NOISE (N2N) (Lehtinen et al., 2018) and CARE (Weigert

| Dataset | Non DL BM3D | Single-image DL DIP | Single-image DL S2S | Multi-image DL N2V | Multi-image DL N2Same | Multi-image DL PN2V | Diversity DL HVAE | Diversity DL DN | Diversity DL HDN | Supervised N2N | Supervised CARE |
|---|---|---|---|---|---|---|---|---|---|---|---|
| Convallaria | 35.45 | - | - | 35.73 | 36.46 | 36.47 | 34.11 | 36.90 | **37.39** | 36.85 | 36.71 |
| Confocal Mice | 37.95 | - | - | 37.56 | 37.96 | 38.17 | 31.62 | 37.82 | **38.28** | 38.19 | 38.39 |
| 2 Photon Mice | 33.81 | - | - | 33.42 | 33.58 | 33.67 | 25.96 | 33.61 | **34.00** | 34.33 | 34.35 |
| Mouse Actin | 33.64 | - | - | 33.39 | 32.55 | 33.86 | 27.24 | 33.99 | **34.12** | 34.60 | 34.20 |
| Mouse Nuclei | 36.20 | - | - | 35.84 | 36.20 | 36.35 | 32.62 | 36.26 | **36.87** | 37.33 | 36.58 |
| Flywing | 23.45 | 24.67 | - | 24.79 | 22.81 | 24.85 | 25.33 | 25.02 | **25.59** | 25.67 | 25.79 |
| BSD68 | 28.56 | 27.96 | 28.61 | 27.70 | 27.95 | 28.46 | 27.20 | 27.42 | **28.82** | 28.86 | 29.07 |
| Set12 | 29.94 | 28.60 | 29.51 | 28.92 | 29.35 | 29.61 | 27.81 | 28.24 | **29.95** | 30.04 | 30.36 |
| BioID Faces | 33.91 | - | - | 32.34 | 34.05 | 33.76 | 31.65 | 33.12 | **34.59** | 35.04 | 35.06 |
| CelebA HQ | 33.28 | - | - | 30.80 | 31.82 | 33.01 | 31.45 | 31.41 | **33.54** | 33.39 | 33.57 |
| Kanji | 20.45 | - | - | 19.95 | 20.28 | 19.40 | 20.44 | 19.47 | **20.72** | 20.56 | 20.64 |
| MNIST | 15.82 | - | - | 19.04 | 18.79 | 13.87 | 20.52 | 19.06 | **20.87** | 20.29 | 20.43 |

Table 1: **Quantitative evaluation of pixel-noise removal on several datasets.** We compare results in terms of mean Peak signal-to-noise ratio (PSNR, in dB) over 5 runs. Note, PSNR is a logarithmic measure. Best performing method is indicated with underline, best performing unsupervised method is shown in bold. Diversity methods allow to generate an arbitrary number of plausible denoised reconstructions given the single observation. For all datasets, our HIERARCHICAL DIVNOISING (HDN) is the new unsupervised SOTA method.

| # latent | PSNR |
|---|---|
| 1 | 27.81 |
| 2 | 28.46 |
| 3 | 28.63 |
| 4 | 28.75 |
| 5 | **28.83** |
| 6 | 28.82 |

| # res. blocks | PSNR |
|---|---|
| 1 | 28.49 |
| 2 | 28.57 |
| 3 | 28.64 |
| 4 | 28.67 |
| 5 | **28.82** |
| 6 | 28.82 |

| Gating | BN | Gen. skips | Free bits | PSNR |
|---|---|---|---|---|
| ✗ | ✓ | ✓ | ✓ | 28.58 |
| ✓ | ✗ | ✓ | ✓ | 28.37 |
| ✓ | ✓ | ✗ | ✓ | 28.61 |
| ✓ | ✓ | ✓ | ✗ | 28.66 |
| ✓ | ✓ | ✓ | ✓ | 28.82 |

(a) Latent variables ablation (5 res. blocks/layer)  (b) Res. blocks ablation (6 latent variables)

(c) Ablation of other units (6 latent layer, 5 res. blocks/layer)

Table 2: **Ablation studies on BSD68 denoising with HDN.** We find that performance usually improves with (a) increasing the number of latent layers, (b) increasing the number of residual blocks per latent layer, (c) incorporating gating blocks, batch norm, skip connections on bottom-up path and free-bits thus validating the importance of each component and validating our careful design. PSNR results (in dB) are shown.

et al., 2018). Note that we are limiting the evaluation of DIP and S2S to 3 and 2 datasets respectively because of their prohibitive training times on full datasets. All training parameters for N2V, PN2V, DN, N2N and CARE are as described in their respective publications (Krull et al., 2019; Prakash et al., 2021; Krull et al., 2020; Goncharova et al., 2020). NOISE2SAME, SELF2SELF and DIP are trained using the default parameters mentioned in Xie et al. (2020), Quan et al. (2020), and Ulyanov et al. (2018), respectively.

**HDN** *Training and Denoising Results.* All but one HDN networks make use of 6 stochastic latent variables $z_1, \ldots, z_6$. The one exception holds for the small *MNIST* images, for which we only use 3 latent variables. For all datasets, training takes between 12 to 24 hours on a single Nvidia P100 GPU requiring only about 6 GB GPU memory. Full training details can be found in Appendix A.1.3 and a detailed description of hardware requirement and training and inference times is reported in Appendix A.9. All denoising results are quantified in terms of Peak-Signal-to-Noise Ratio (PSNR) against available ground truth (GT) in Table 1. PSNR is a logarithmic measure and must be interpreted as such. Given a noisy image, HDN can generate an arbitrary number of plausible denoised images. We therefore approximated the conditional expected mean (*i.e.*, MMSE estimate) by averaging 100 independently generated denoised samples per noisy input. Fig. 1 shows qualitative denoising results.

For datasets with higher noise and hence more ambiguity such as *DenoiSeg Flywing*, the generated samples are more diverse, accounting for the uncertainty in the noisy input. Note also that each generated denoised sample appears less smooth than the MMSE estimate (see *e.g.* BSD68 or BioID results in Fig. 1). Most importantly, HDN outperforms all unsupervised baselines on all datasets and even supersedes the supervised baselines on 3 occasions. A big advantage of a fully unsupervised approach like HDN is that it does not require any paired data for training. Additional results using the SSIM metric confirming this trend and are given in Appendix A.5.

*Practical utility of diverse denoised samples from* **HDN.** Additional to the ability to generate diverse denoised samples, we can aggregate these samples into point estimates such as the pixel-wise median (Appendix A.6) or MAP estimates (Appendix A.7). Additionally, the diversity of HDN

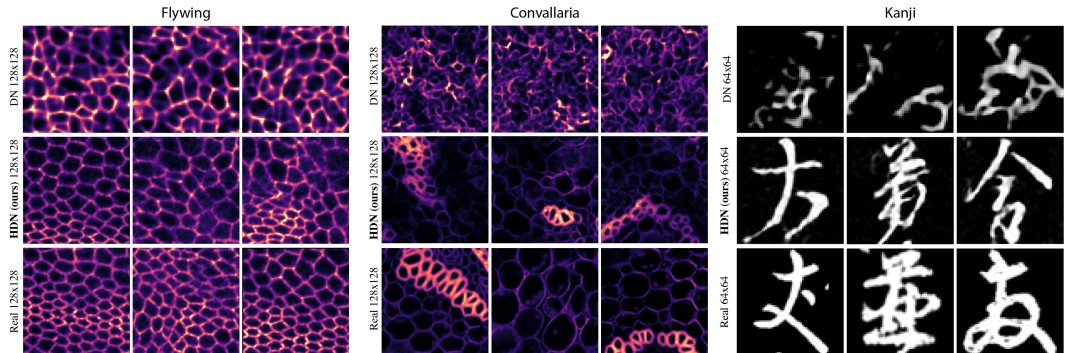

Figure 2: **Accuracy of the HDN generative model.** We show real images *(bottom row)*, unconditionally generated images from HIERARCHICAL DIVNOISING *(middle row)* and those from DIVNOISING *(top row)* at different resolutions for the *DenoiSeg Flywing*, *FU-PN2V Convallaria* and Kanji datasets. The generated images from our method look much more realistic compared to those generated by DIVNOISING, thereby validating the fact that our method is much more expressive and powerful compared to DIVNOISING.

| | Unsupervised | | | | | | Supervised |
|---|---|---|---|---|---|---|---|
| | N2V | PN2V | Struct N2V | DN | HDN (Ours) | HDN$_{3-6}$ (Ours) | CARE |
| Struct.Convallria | 29.33 | 29.43 | 30.02 | 31.09 | 29.96 | **31.36** | 31.56 |

Table 3: **Quantitative striping artefacts removal.** On the real-world microscopy data from Broaddus et al. (2020) (Struct. Convallaria), the proposed HDN$_{3-6}$ produces state-of-the-art results in terms of PSNR (in dB), outperforming all unsupervised baselines. Note that PSNR is a logarithmic measure.

samples is indicative of the uncertainty in the raw data, with more diverse samples pointing at a greater uncertainty in the input images (see Appendix A.12). On top of all this, in Appendix A.8 we showcase a practical example of an instance segmentation task which greatly benefits from diverse samples being available.

***Comparison of* HDN *with Vanilla Hierarchical* VAE *(HVAE) and* DN.** Here, we compare the performance of HDN with Vanilla HVAE and DN, conceptually the two closest baselines. Vanilla HVAE can be used for denoising and does not need a noise model. We employed a HVAE with the same architecture as HDN and observe (see Table 1) that HDN is much better for all 12 datasets.

Arguably, the superior performance of HDN can be attributed to the explicit use of a noise model. However, Table 1 also shows that HDN is leading to much improved results than DN, indicating that the noise model is not the only factor. The carefully designed architecture of HDN is contributing by learning a posterior of clean images that better fits the true clean data distribution than the one learned by DN (see also Fig. 2). A detailed description of the architectural differences between DN and HDN are provided in Appendix A.11.

The ablation studies presented in Table 2 additionally demonstrate that all architectural components of HDN contribute to the overall performance. Also note that the careful design of these architectural blocks allows HDN to yield superior performance while still being computationally efficient (requiring only around 6 GB on a common Tesla P100 GPU, training for only about 24 hours). Other hierarchical VAE models proposed for SOTA image generation (Vahdat & Kautz, 2020; Child, 2021) need to be trained for days or even weeks on multiple GPUs.

***Accuracy of the* HDN *generative model.*** Since both HDN and DN are fully convolutional, we can also use them for arbitrary size unconditioned image generation (other than denoising). For the DN setup, we sample from the unit normal Gaussian prior and for HDN, we sample from the learned priors as discussed in Section 4 and then generate results using the *top-down* network. Note that in either case, the images are generated by sampling the latent code from the prior without conditioning on an input image. We show a random selection of generated images for *DenoiSeg Flywing*, *FU-PN2V Convallaria* and *Kanji* datasets in Fig. 2 (and more results in Appendix A.2). When comparing generated images from both setups with real images from the respective datasets, it is rather evident that HDN has learned a fairly accurate posterior of clean images, while DN has failed to do so.

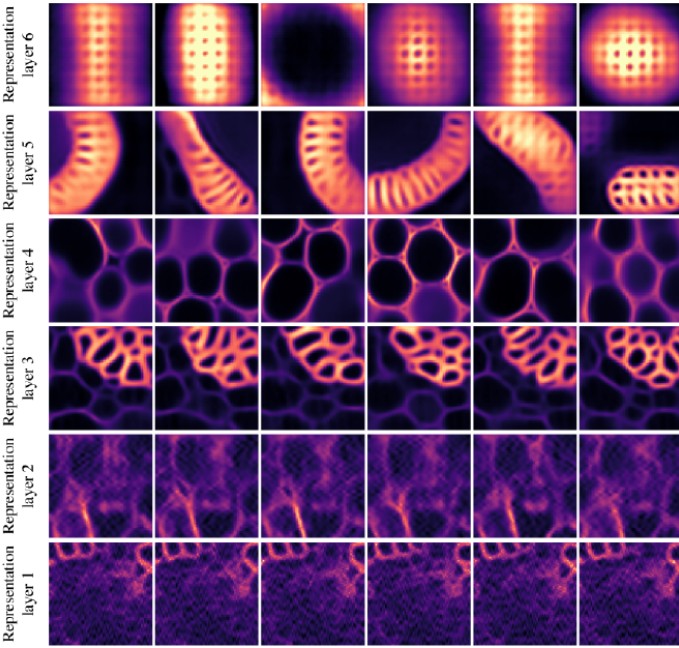

Figure 3: **Visualizing what HIERARCHICAL DIVNOISING encodes at different latent layers.** Here we inspect the contribution of each latent layer in our $n = 6$ latent layer hierarchy when trained with data containing structured noise. To inspect contribution of latent variable $\mathbf{z}_i$, we fix the latent variables of layers $j > i$, draw $k = 6$ conditional samples for layer $i$, and take the mean of the conditional distributions at layers $m < i$ generating then a total of $k$ denoised samples (columns). This is repeated for every latent variable layer (rows). Please see Appendix A.4 for details. An interesting observation is that structures that resemble the structured line artefacts are only visible in layers 1 and 2, thus motivating the proposed $\text{HDN}_{3-6}$ network (see Tables 3 and 4).

## 6 INTERPRETABLE STRUCTURED NOISE REMOVAL WITH HDN

HDN is using a pixel noise model that does not capture any property of structured noises (artefacts). Here we show and analyze the ability of HDN to remove structured noise in different imaging modalities, even if artefacts affect many pixels in correlated ways. To our knowledge, we are the first to show this, additionally looking at it through the lens of interpretability, without utilizing prior knowledge about structured noise and without learning/employing a structured noise model.

Additional to pixel-noise, microscopy images can be subject to certain artefacts (see Fig. 4 and Appendix 6 for a detailed characterization of such artefacts). Hence, we applied the HDN setup described in Section 4 with 6 stochastic latent layers on such a confocal microscopy dataset from Broaddus et al. (2020), subject to intrinsic pixel-noise as well as said artefacts. In order to apply HDN, we learnt a pixel-wise GMM noise model as mentioned in Section 4. We observed that HDN did not remove these striping patterns (see Appendix Fig. 11 and Table 3), presumably because the hierarchical latent space of HDN is expressive enough to encode all real image features and artefacts contained in the training data.

However, the hierarchical nature of HDN lends an interpretable view of the contributions of individual latent variables $\mathbf{z}_1, \ldots, \mathbf{z}_n$ to the restored image. We use the trained HDN network to visualize the image aspects captured at hierarchy levels 1 to $n$ in Fig. 3. Somewhat surprisingly, we find that striping artefacts are almost exclusively captured in $\mathbf{z}_1$ and $\mathbf{z}_2$, the two bottom-most levels of the hierarchy and corresponding to the highest resolution image features. Motivated by this finding, we modified our HDN setup to compute the conditional distribution $q_{\phi,\theta}(\mathbf{z}_i | \mathbf{z}_{j>i}, \mathbf{x})$ of Eq. 10 to only use information from the *top-down* pass, neglecting the encoded input that is usually received from *bottom-up* computations.

Since we want to exclude artefacts which are captured in $\mathbf{z}_1$ and $\mathbf{z}_2$ we apply these modifications only to these two layers and keep all operations for layers 3 to 6 unchanged. We denote this particular setup with $\text{HDN}_{3-6}$, listing all layers that operate without alteration. As can be seen in Table 3, the trained $\text{HDN}_{3-6}$ network outperforms all other unsupervised methods by a rather impressive margin

|      | HDN   | HDN$_{2-6}$ | HDN$_{3-6}$ | HDN$_{4-6}$ |
|------|-------|-------------|-------------|-------------|
| PSNR | 29.96 | 31.24       | **31.36**   | 28.13       |

Table 4: **Performance of HDN networks with "deactivated" latent layers.** We show results for HDN setups (in terms of PSNR in dB) with different layers of the original HDN architecture "deactivated" (see main text and Appendix A.4 for more details). In line with our observations in Fig. 3, the HDN$_{3-6}$ setup leads to the best performance on the data of Broaddus et al. (2020), which we used here.

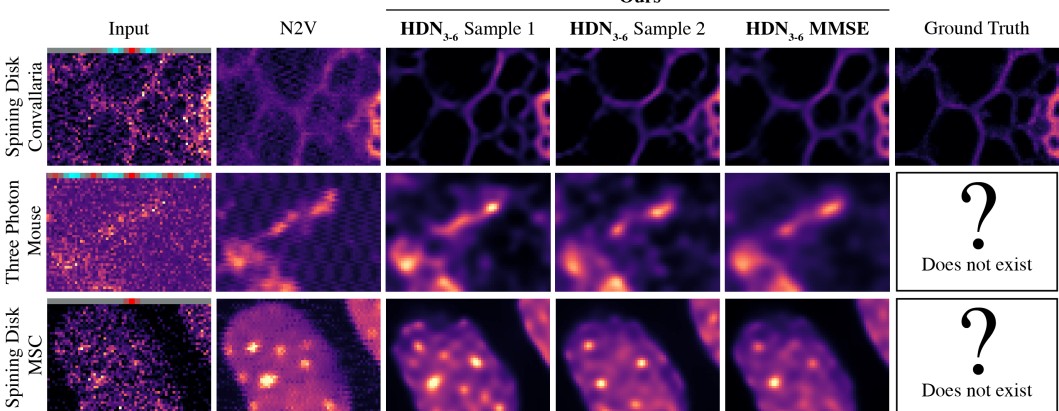

Figure 4: **Qualitative structured noise results on microscopy datasets.** For three microscopy datasets coming from different modalities *(rows)*, we show artefact removal results for one representative crop of input images (leftmost column). We illustrate the nature of the structured artefacts for each dataset by indicating the spatial autocorrelation of the noise on top of the raw images in the first column. See Appendix 6 for a more detailed description. Note that N2V recovers structured noise and fails to remove it. We show two randomly chosen HDN$_{3-6}$ samples which successfully remove structured noise, the HDN$_{3-6}$ MMSE estimate derived by averaging 100 restored samples and the ground truth image.

and is now removing the undesired artefacts (see the third, fourth and fifth columns in first row in Fig. 4 for this dataset). To validate our design further, we also computed results by "deactivating" only the bottom-most layer (HDN$_{2-6}$), and the bottom-most three layers (HDN$_{4-6}$). All results are shown in Table 4. The best results are obtained, as expected, with HDN$_{3-6}$.

Furthermore, we demonstrate the applicability of our proposed HDN$_{3-6}$ setup for removing microscopy artefacts for two more real-world datasets: $(i)$ mouse cells imaged with three-photon microscopy and $(ii)$ mesenchymal stem cells (MSC) imaged with a spinning disk confocal microscope. We present the qualitative results for all three datasets in Fig. 4. Note that ground truth for the spinning disk MSC images was not acquired and for the three-photon data is technically unobtainable, illustrating how important our unsupervised HDN$_{3-6}$ setup is for microscopy image data analysis.

# 7 CONCLUSIONS AND FUTURE WORK

In this work, we have looked at unsupervised image denoising and artefact removal with emphasis on generating diverse restorations that can efficiently be sampled from a posterior of clean images, learned from noisy observations. We first introduced HIERARCHICAL DIVNOISING (HDN), a new and carefully designed network utilizing expressive hierarchical latent spaces and leading to SOTA results on 12 unsupervised denoising benchmarks. Additionally, we showed that the representations learned by different latent layers of HDN are interpretable, allowing us to leverage these insights to devise modified setups for structured noise removal in real-world microscopy datasets.

Selective "deactivation" of latent layers will find a plethora of practical applications. Microscopy data is typically diffraction limited, and the pixel-resolution of micrographs typically exceeds the optical resolution of true sample structures. In other words, true image signal is blurred by a point spread function (PSF), but imaging artefacts are often not. Hence, the spatial scale of the true signal in microscopy data is not the same as the spatial scale of many structured microscopy noises and the method we proposed will likely lead to excellent results.

Our work takes an interpretability-first perspective in the context of artefact removal. While, as we have shown, our method leads to SOTA results for pixel-wise noise removal and for structured noise removal in microscopy, the situation is more complicated for structured noises in other data domains (see Appendix A.10 for details) and is slated for future work.

**ACKNOWLEDGEMENTS.** The authors would like to thank Alexander Krull at University of Birmingham and Sander Dieleman at DeepMind for insightful discussions, the Prevedel Lab at EMBL Heidelberg for helpful discussions and providing the Three Photon Mouse dataset and the Tabler Lab at MPI-CBG for providing the Spinning Disc MSC dataset. Funding was provided from the core budget of the Max-Planck Institute of Molecular Cell Biology and Genetics (MPI-CBG), the MPI for Physics of Complex Systems, the Human Technopole, and the BMBF under code 031L0102 (de.NBI), and the DFG under code JU3110/1-1 (FiSS). We also want to thank the Scientific Computing Facility at MPI-CBG for giving us access to their computing resources.

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

# A Appendix

## A.1 Datasets and Training Details

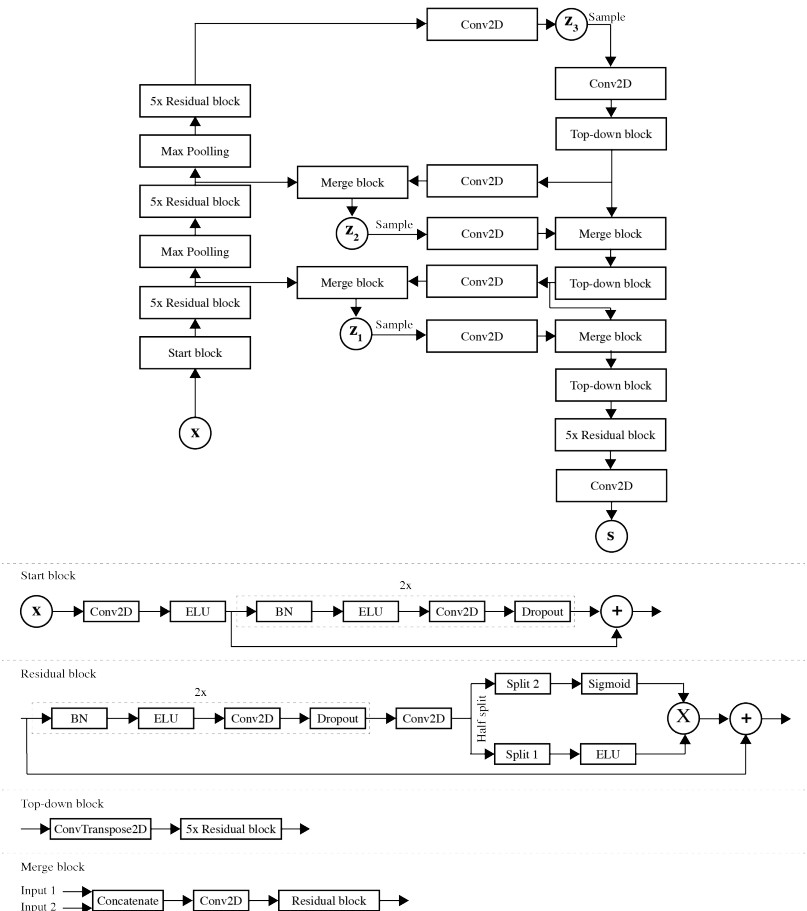

Figure 5: **Hierarchical DivNoising (HDN) network architecture.** The network architecture used for *MNIST* dataset is shown. For all other datasets, 6 stochastic latent groups are used instead of the 3 groups shown here.

### A.1.1 Datasets Corrupted with Pixel-wise Noise

We consider six microscopy datasets namely, the $(i)$ *FU-PN2V Convallaria* dataset from Krull et al. (2020); Prakash et al. (2019b) which shows $1024 \times 1024$ images of a intrinsically real-world noisy *Convallaria* sections, the $(ii)$ *FU-PN2V Mouse Nuclei* dataset from Prakash et al. (2019b), showing $512 \times 512$ images of a mouse skull nuclei, $(iii)$ *FU-PN2V Mouse Actin* dataset from Prakash et al. (2019b), showing $1024 \times 1024$ images of actin labeled sample, $(iv)$ *Two Photon Mice* dataset from Zhang et al. (2019) of size $512 \times 512$, $(v)$ *Confocal Mice* dataset from Zhang et al. (2019) of size $512 \times 512$, and $(vi)$ *DenoiSeg Flywing* dataset from Buchholz et al. (2020), showing $512 \times 512$ images of a developing Flywing. Following Prakash et al. (2021), we synthetically corrupted *DenoiSeg Flywing* images with zero mean Gaussian noise of standard deviation 70 while the other five microscopy datasets are real-world intrinsically noisy images. The *Two Photon Mice* and *Confocal Mice* datasets are available in multiple noise levels starting from highest noise level to the lowest noise level. We present results for the highest noise level (most noisy images) for both these datasets.

The train and validation splits for these datasets are as described in the respective publication. For the *FU-PN2V Convallaria* dataset, the test set consists of 100 images of size $512 \times 512$, for the *FU-PN2V*

*Mouse Actin* dataset, the test set consists of 100 images of size $1024 \times 512$, for the *FU-PN2V Mouse Nuclei* dataset the test set has 200 images of size $512 \times 256$ while the test set for *DenoiSeg Flywing* dataset consists of 42 images of size $512 \times 512$. Both *Two Photon Mice* and *Confocal Mice* datasets consist of 50 test images each of size $512 \times 512$.

From the domain of natural images, we consider two grayscale test datasets namely, $(vii)$ the popular *BSD68* dataset from (Roth & Black, 2005) containing 68 natural images of different sizes, and $(viii)$ the *Set12* dataset from (Zhang et al., 2017) containing 12 images of either size $512 \times 512$ or $256 \times 256$. Both datasets have been synthetically corrupted with zero mean Gaussian noise of standard deviation 25. For training and validation, we use a set of 400 natural images of size $180 \times 180$ as released by Zhang et al. (2017) and has later also been used in Krull et al. (2019).

Next, we chose to use $(ix)$ the popular *MNIST* dataset (LeCun et al., 1998) showing $28 \times 28$ images of digits, and $(x)$ the *Kuzushiji-Kanji* dataset (Clanuwat et al., 2018) showing $64 \times 64$ images of Kanji characters. Both datasets have been synthetically corrupted with zero mean Gaussian noise of standard deviation 140. For the Kanji dataset, we randomly select 119360 images for training and 14042 images for validation. The test set contains 100 randomly selected images not present in either training or validation sets. For the *MNIST* dataset, the publicly available data has two splits containing 60000 training images and 10000 test images. We further divide the training images into training and validation sets containing 54000 and 6000 images, respectively. We take a random subset of 100 images from the test data and use this as our test set to evaluate all methods. The noisy train, validation and test images for *Kanji* and *MNIST* data will be released publicly.

Finally, we selected two datasets of faces, namely, $(xi)$ the *BioID Faces* dataset (noa) showing $286 \times 384$ images of different persons under different lighting conditions, and $(xii)$ the *CelebA HQ* dataset at $256 \times 256$ image resolution containing faces of celebrities from Karras et al. (2017)[2]. The CelebA HQ dataset has RGB images and we take only the red channel images for our experiments. Both facial datasets have been synthetically corrupted with zero mean Gaussian noise of standard deviation 15. For the *BioID Faces* dataset, we choose 340 images in the training set and 60 images in the validation set while the test set contains 100 additional images. For the *CelebA HQ* dataset, the train and validation sets contain 27000 and 1500 images, respectively, while the test set contains 100 additional images. We will release the noisy versions of these datasets publicly.

### A.1.2 DATASETS CORRUPTED WITH STRUCTURED NOISE/ARTEFACTS

We consider three real-world intrinsically noisy microscopy datasets namely, the $(i)$ *Struct. Convallaria* dataset from Broaddus et al. (2020) which shows $1024 \times 1024$ images of a intrinsically real-world noisy *Convallaria* sections imaged with spinning disk confocal microscope, the $(ii)$ *Three Photon Mouse* dataset showing intravital calcium imaging data ($256 \times 256$ images) of astrocytes expressing GCaMP6s, acquired in the cortex of an anesthesized mouse at 814µm depth below the brain surface with three photon microscopy, and $(iii)$ *Spinning disk MSC* dataset showing $1024 \times 1024$ images of mouse Mesenchymal Stem Cells imaged with a spinning disk confocal microscope.

For the *Struct. Convallaria* dataset, the test set consists of 100 images of size $512 \times 512$, for the *Three Photon Mouse* dataset, the test set consists of 500 images of size $256 \times 256$ while for the *MSC* dataset the test set has 100 images of size $512 \times 512$. All these datasets are intrinsically noisy containing both pixel noise and structured noise/artefacts.

***Description of structured noise/artefacts in microscopy images.*** Structured noise/artefacts are common in microscopy and arise due to various reasons (*e.g.*camera/detector imperfections). However, spatial correlations of such artefacts are quite variable, rendering unsupervised artefact removal nontrivial. To better characterize structured noises in our datasets, we compute the spatial autocorrelation of the noise (See Appendix Fig. 6).

For the Convallaria dataset subject to structured noise, we obtain the residual noise image by subtracting the ground truth from the raw image data. Since for the three photon mouse and spinning disk MSC datasets no ground truth images are available, we estimate the noise at background areas (empty areas) of the data, assuming that such regions only contain noise. For reference, we also show the spatial autocorrelation for Gaussian noise (non-structured pixel-wise noise) in the first column

---

[2]The full dataset can be downloaded from the Google Drive link at `https://github.com/suvojit-0x55aa/celebA-HQ-dataset-download`

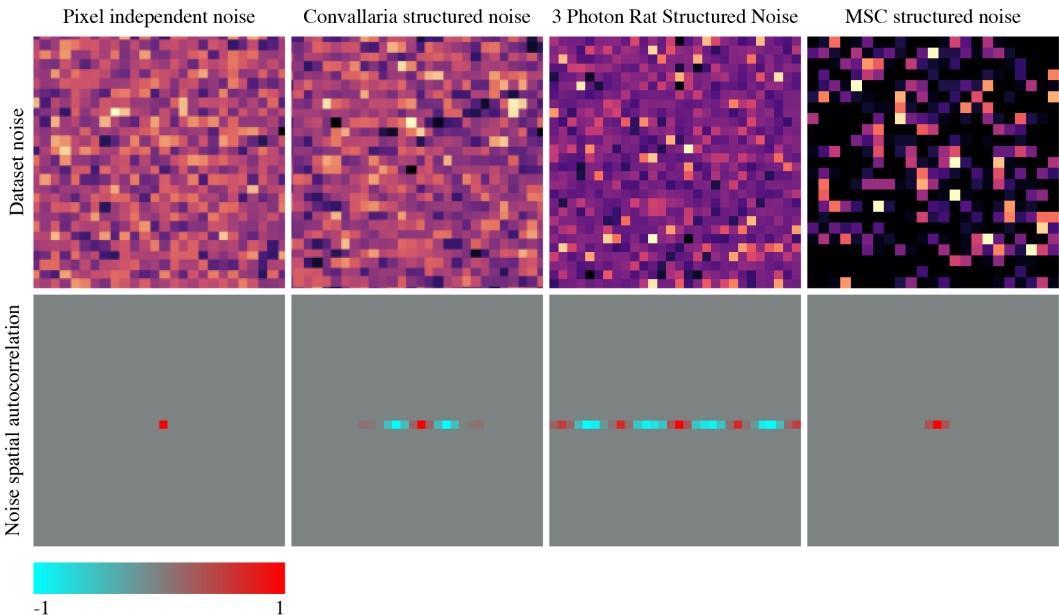

Figure 6: **Illustration of structured noise/artefacts in microscopy images.** To better illustrate the structured noise in the considered datasets, we take the noise (first row) and compute their spatial autocorrelation (second row). For structured Convallaria dataset, the noise is obtained by subtracting the ground truth image with noisy raw image. For the three photon mouse and spinning disk MSC datasets, ground truth images are not available and hence, the noise is estimated at background regions (empty areas) of the images, which supposedly only contain noise. For reference, we also show the spatial autocorrelation of Gaussian noise (pixel independent noise) in first column. For each case, we show the autocorrelation of noise for one pixel in the center (shown in red). For all three real microscopy datasets, there is a spatial correlation between noise unlike the Gaussian noise case where the noise is independent per pixel. Also note that the spatial extent of the structured noise is different for different datasets. Still, our proposed HDN$_{3-6}$ makes no assumption on the prior knowledge of spatial extent of these artefacts and successfully removes them for all considered datasets whereas other methods such as NOISE2VOID only remove pixel noise and recover structured noise (see Fig. 4 in main text).

of Appendix Fig. 6. It can be seen that for all real-world microscopy datasets the noise is spatially correlated unlike pixel-wise Gaussian noise.

### A.1.3 TRAINING DETAILS OF HIERARCHICAL DIVNOISING

Our HDN network is fully convolutional and consists of multiple hierarchical stochastic latent variables. For our experiments we use a network with 6 hierarchical latent variables, only for the *MNIST* dataset we have only used a hierarchy of height 3. Each latent group has $H_i \times W_i \times 32$ dimensions where $H_i$ and $W_i$ represent the dimensionality of the feature maps at layer $i$. The initial number of filters is set to 64 and we utilize batch-normalization (Ioffe & Szegedy, 2015) and dropout (Srivastava et al., 2014) with a probability of 0.2. Both *top-down* and *bottom-up* networks have 5 residual blocks between each latent group. A schematic of the residual blocks and the full network architecture is shown in Fig. 5.

We use an initial learning rate of 0.0003 and always train for 200000 steps using Adamax (Kingma & Ba, 2015). During training, we extract random patches from the training data ($128 \times 128$ patches for *BioID Faces* and natural image datasets, $256 \times 256$ patches for *CelebA HQ*, $28 \times 28$ patches for *MNIST*, and $64 \times 64$ patches for all other datasets). A batch size of 16 was used for *BioID Faces* and *BSDD68* while a batch size of 4 was used for the *CelebA HQ* dataset . For all other datasets, a batch size of 64 was used. To prevent *KL vanishing* (Bowman et al., 2015), we use the *free bits* approach as described in Kingma et al. (2016); Chen et al. (2016). For experiments on all datasets other than the *MNIST* dataset, we set the value of *free bits* to 1.0, while for the *MNIST* dataset a value of 0.5 was used.

### A.1.4 Architecture of Hierarchical DivNoising

Figure 5 shows our Hierarchical DivNoising architecture with 3 stochastic latent variables, as used for experiments on the *MNIST* data. For all other experiments our architecture is the same, but has 6 latent variable groups instead of 3. Please note that we have used the exact same architecture for pixel-wise and structured noise removal experiments. Our networks with 6 latent groups need around 6 GB of GPU memory and were trained on a Tesla P100 GPU. The training time until convergence varies from roughly 12 hours to 1 day depending on the dataset.

## A.2 How Accurate is the Hierarchical DivNoising posterior?

In Section 5 in main text, we qualitatively compared the learned posterior of Hierarchical DivNoising with DivNoising. Since both methods are fully convolutional, we can generate images of arbitrary size. Expanding on what was discussed in Section 5 of main text, here we show more generated images for multiple datasets. For the DivNoising setup, we sample from the unit normal Gaussian prior and for Hierarchical DivNoising, we sample from the learned priors as discussed in Section 3 and then generate results using the *top-down* network. Note that in either case, the images are generated by sampling the latent code from the prior without conditioning on an input image. Below, we show a random selection of generated images for *DenoiSeg Flywing*, *FU-PN2V Convallaria*, *Kanji*, *MNIST* and *natural images* datasets in Fig. 7, Fig. 8, Fig. 9 and Fig. 10, respectively at different image resolutions.

When comparing generated images from both setups with real images from the respective datasets, it is rather evident that Hierarchical DivNoising learns a much more accurate posterior of clean images than DivNoising. For the *MNIST*, *Kanji*, *FU-PN2V Convallaria* and *DenoiSeg Flywing* datasets, the images generated by Hierarchical DivNoising have a very realistic appearance. For the *natural image* dataset the situation is rather different. Generated natural images do not have the same quality as instances of the training dataset do, offering lots of room for future improvement. Hence, it appears that learning a suitable latent space encoding of clean natural images is much harder than for the other datasets.

## A.3 Qualitative Results for Structured Noise Experiments with HDN

Following what was discussed in Section 6, here we show qualitative results for microscopy striping artefacts removal experiments for HDN (Fig. 11). Clearly, HDN is too expressive and captures the undesirable striping patterns and fails to remove them. Both the individual samples from HDN and the MMSE estimate (averaging 100 HDN samples) clearly show striping patterns. Owing to this, the restoration performance of HDN is rather poor (see Table 3). Hence, we have proposed $HDN_{3-6}$ to overcome this drawback of HDN (see Section 6 and Table 3 in main text for more details) which establishes a new state-of-the-art.

## A.4 Visualizing Patterns Encoded by HDN Latent Layers and Targeted Deactivation

In this section we present more details regarding Fig. 3 in main text pertaining to visualizing what the individual levels of a HDN learned to represent when trained on *Struct. Convallaria* dataset containing structured noise. The idea for this way of inspecting a Ladder VAE is not new and can also be found in a GitHub repo[3] authored by a member of the team around Liévin et al. (2019).

More specifically, we want to visualize what our 6-layered HDN model has learned in Section 6 of the main text. We inspect layer $i$ by performing the following steps:

- We draw a sample from all latent variables for layers $j > i$.
- We then draw 6 samples from layer $i$ (conditioned on the samples we drew just before).
- For each of these 6 samples, we take the mean of the conditional distributions at layers $m < i$, *i.e.* we do not sample for these layers.
- We generate a total of 6 images given the latent variable samples and means from above.

---

[3] https://github.com/addtt/ladder-vae-pytorch

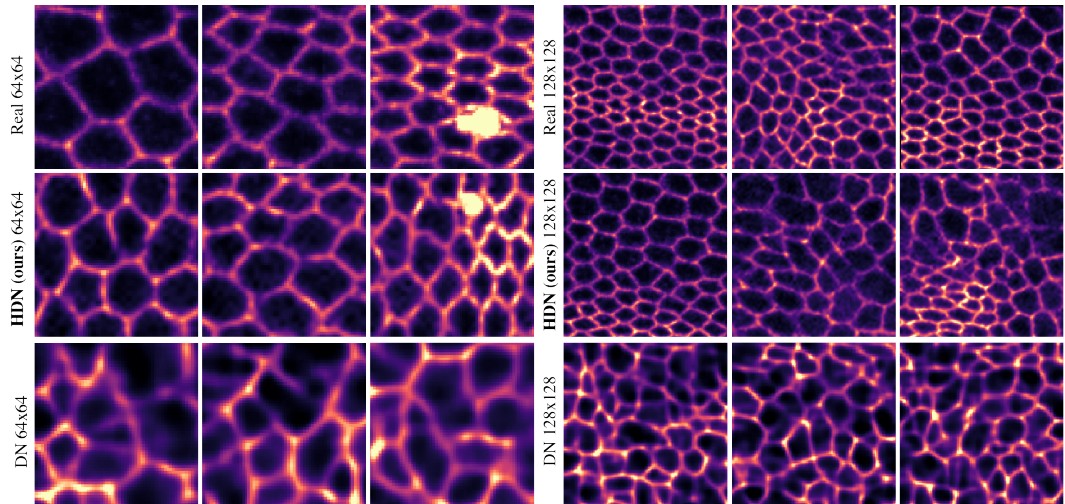

Figure 7: **Accuracy of the learned posterior.** We show real images *(top row)*, generated images from HIERARCHICAL DIVNOISING *(middle row)* and generated images from DIVNOISING *(bottom row)* at different resolutions for the *DenoiSeg Flywing* dataset. The images from our method look much more realistic compared to those generated by DIVNOISING, thereby validating the fact that our method learns a much better posterior distribution of clean images compared to DIVNOISING.

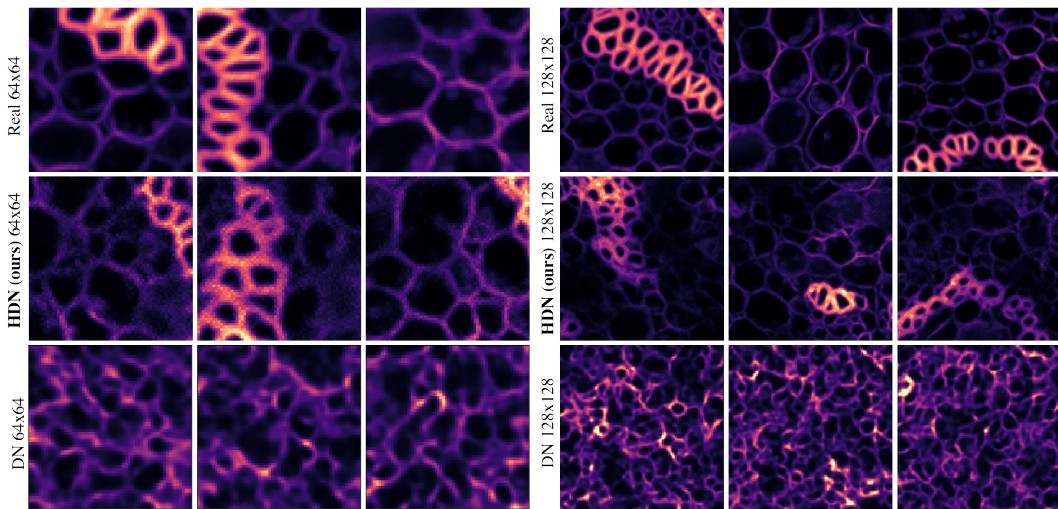

Figure 8: **Accuracy of the learned posterior.** We show real images *(top row)*, generated images from HIERARCHICAL DIVNOISING *(middle row)* and generated images from DIVNOISING *(bottom row)* at different resolutions for the *FU-PN2V Convallaria* dataset. The images from our method look much more realistic compared to those generated by DIVNOISING, thereby validating the fact that our method learns a much better posterior distribution of clean images compared to DIVNOISING.

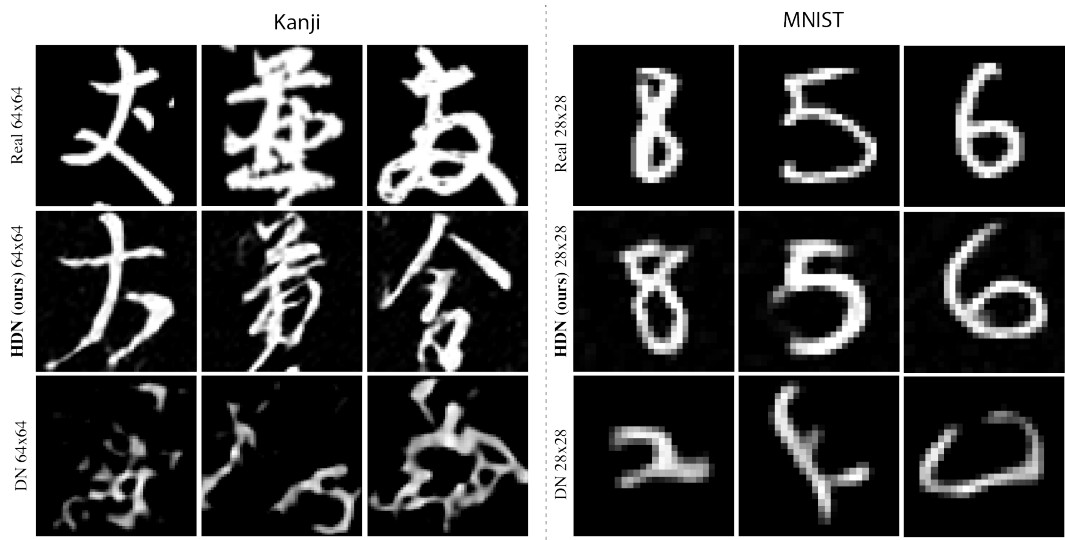

Figure 9: **Accuracy of the learned posterior.** We show real images *(top row)*, generated images from HIERARCHICAL DIVNOISING *(middle row)* and generated images from DIVNOISING *(bottom row)* for the *Kanji* dataset in first half and the *MNIST* dataset in second half. The images from our method look much more realistic compared to those generated by DIVNOISING, thereby validating the fact that our method learns a much better posterior distribution of clean images compared to DIVNOISING.

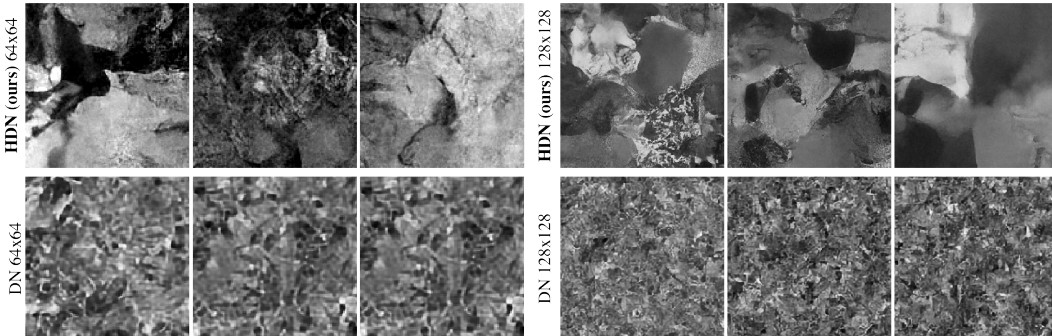

Figure 10: **Accuracy of the learned posterior.** We show generated images from HIERARCHICAL DIVNOISING *(top row)* and generated images from DIVNOISING *(bottom row)* at different resolutions for *natural images*. The images from our method look much more realistic compared to those generated by DIVNOISING, although there seems to be plenty of room for improvement. The generated images by HDN vaguely resemble structures such as rock, clouds, etc. More meaningful structures like people, animals, etc. are not generated at all. This indicates that our method is not powerful enough to capture the tremendous diversity in natural image datasets but still gives SOTA denoising performance on natural image benchmarks (see Table 1 in main text and Appendix Table 5).

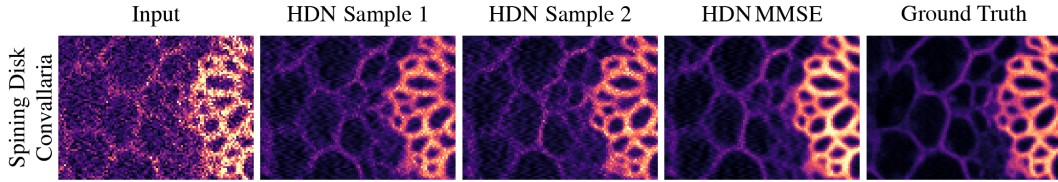

Figure 11: **Qualitative results of structured noise experiments with HDN.** As explained in the main text, HDN owing to its high expressivity captures striping artefacts and fails to remove them, thereby leading to inferior restoration performance (see Table 3). Hence, we propose the modified HDN$_{3-6}$ for microscopy structured noise removal (see Fig. 4 and Table 3 in main text) which establishes a new state-of-the-art.

The results of this procedure on the HDN network used in Section 6 can be seen in Fig. 3 in main text.

By visually inspecting Fig. 3 in main text one can observe that the bottom two layers (layers 1 and 2) learned fine details, including the line artefacts we would like to remove from input images. The other layers (layers 3 to 6), on the other hand, either learn smoother global structures or learn such abstracted concepts that the used visualization method is not very informative.

Still, the created visuals help us to understand why the full HDN leads to worse results for removing microscopy line artefacts than a DN setup did (see Table 3 in main text). Owing to the high expressivity of the HDN model, the structured noise can actually be encoded in the hierarchical latent spaces and the artefacts will therefore be faithfully be represented in the otherwise denoised resulting image.

Motivated by these observations, we proposed a simple modification to HDN that has then led to SOTA results on the given benchmark dataset (see again Table 3 in the main text). We showed that by "deactivating" layers 1 and 2, *i.e.* the ones that show artefact-like patterns in Fig. 3, the modified HDN model restored the data well and removed pixel noises as well as the undesired line artefacts (structured noise).

We also computed results by "deactivating" only the bottom-most layer (HDN$_{2-6}$), the lowest two layers as described in the main text (HDN$_{3-6}$), and the bottom-most three layers (HDN$_{4-6}$). All results are shown in terms of peak signal-to-noise ratio (PSNR) in Table 4 in main text. The best results are obtained with HDN$_{3-6}$ setup.

### A.5   PIXEL-WISE DENOISING IN TERMS OF SSIM

While we have measured pixel-denoising performance in the main text in terms of PSNR, here we quantify results also in terms of Structural Similarity (SSIM), see Table 5. Our proposed HDN setup outperforms all unsupervised baselines for all datasets except for Set12 dataset where it is the second best. Notably, HDN even surpasses SSIM results of supervised methods on 4 datasets.

### A.6   PIXEL-WISE MEDIAN ESTIMATES

We compute the pixel-wise median estimates (computed by taking the per-pixel median for 100 denoised samples from HDN) and compare them to MMSE estimates in terms of PSNR (dB). The results are reported in Table 6. MMSE estimates are marginally better than pixel-wise median estimates for all datasets.

### A.7   MAP ESTIMATES

We show the utility of having diverse denoised samples by computing MAP estimates and qualitatively comparing them to MMSE estimates. We present the case for the MNIST dataset (LeCun et al., 1998) (Fig. 12) and the *eBook* dataset (Marsh, 2004) (Fig. 13), both also used in Prakash et al. (2021).

In order to compute the MAP estimate, we follow the procedure of Prakash et al. (2021) involving a mean-shift clustering approach with 100 ddiverse HDN samples. The MMSE estimates represent

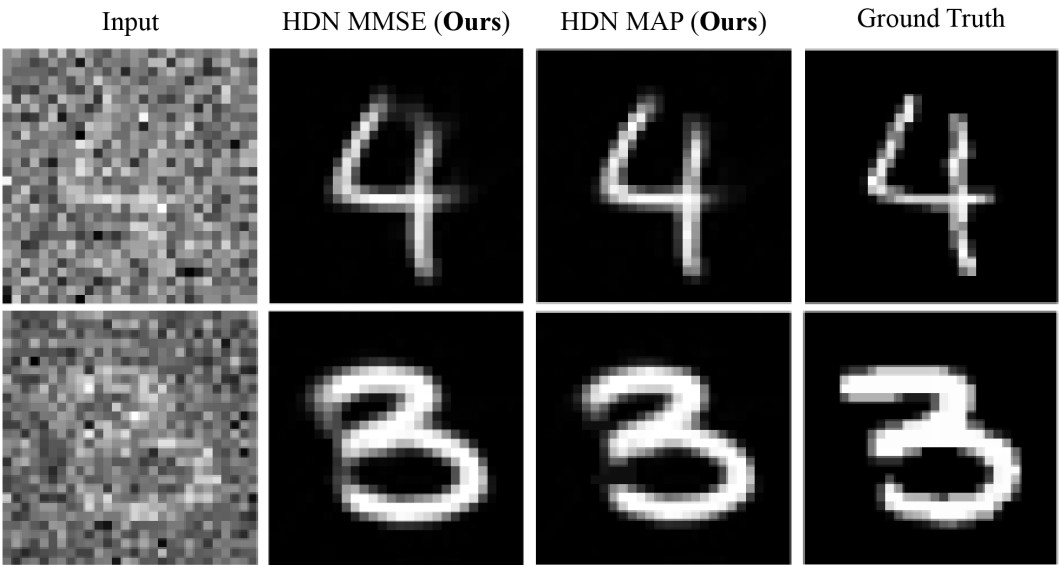

Figure 12: **Comparison of HDN MMSE and MAP estimates for two representative MNIST inputs.** We show noisy input images *(first column)*, HIERARCHICAL DIVNOISING MMSE estimates computed from 100 samples *(second column)*, HIERARCHICAL DIVNOISING MAP estimates *(third column)* computed using a mean-shift clustering approach with the same 100 samples and ground truth images *(fourth column)*. Note that the MMSE estimate is a compromise between all possible denoised solutions and hence the obtained MMSE results have faint overlays suggesting a compromise between the digits 9 and 4 (first row) and the digits 3 and 8 (second row) while the MAP estimate is committing to one interpretation.

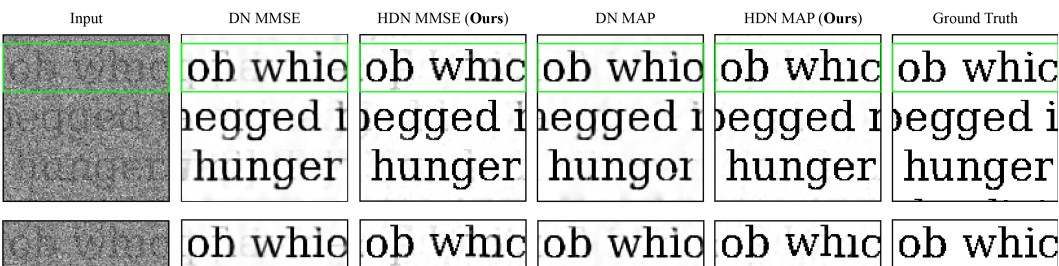

Figure 13: **Comparison of HDN MMSE and MAP estimates for eBook dataset.** We show noisy input image *(first column)*, DIVNOISING MMSE estimate taken from Prakash et al. (2021) *(second column)*, our HIERARCHICAL DIVNOISING MMSE estimates computed from 100 samples *(third column)*, DIVNOISING MAP estimate taken from Prakash et al. (2021) *(fourth column)*, our HIERARCHICAL DIVNOISING estimate *(fifth column)* and ground truth image *(sixth column)*. Note that the MMSE estimate is a compromise between all possible denoised solutions and hence the obtained MMSE results have faint overlays suggesting a compromise between different possible letters while the MAP estimate commits to one interpretation of letters.

| Dataset | Non DL | Single-image DL | | Multi-image DL | | | Diversity DL | | | Supervised | |
|---|---|---|---|---|---|---|---|---|---|---|---|
| | **BM3D** | **DIP** | **S2S** | **N2V** | **N2Same** | **PN2V** | **HVAE** | **DN** | **HDN** | **N2N** | **CARE** |
| | | | | | SSIM | | | | | | |
| Convallaria | 0.939 | - | - | 0.954 | 0.963 | 0.959 | 0.911 | **0.966** | **0.966** | 0.963 | 0.962 |
| Confocal Mice | 0.963 | - | - | 0.963 | 0.965 | 0.965 | 0.693 | 0.964 | **0.966** | 0.966 | 0.967 |
| 2 Photon Mice | **0.924** | - | - | 0.919 | 0.918 | 0.920 | 0.355 | 0.918 | 0.920 | 0.924 | 0.906 |
| Mouse Actin | 0.855 | - | - | 0.850 | 0.818 | 0.867 | 0.653 | 0.877 | **0.889** | 0.890 | 0.882 |
| Mouse Nuclei | 0.952 | - | - | 0.952 | 0.958 | 0.955 | 0.852 | 0.958 | **0.962** | 0.967 | 0.959 |
| Flywing | 0.721 | - | - | 0.843 | 0.482 | 0.821 | 0.850 | 0.842 | **0.852** | 0.865 | 0.866 |
| BSD68 | 0.801 | 0.774 | 0.803 | 0.778 | 0.776 | 0.801 | 0.764 | 0.754 | **0.812** | 0.812 | 0.827 |
| Set12 | **0.850** | 0.772 | 0.832 | 0.828 | 0.818 | 0.831 | 0.800 | 0.787 | 0.837 | 0.844 | 0.860 |
| BioID Faces | 0.909 | - | - | 0.865 | 0.907 | 0.891 | 0.770 | 0.886 | **0.920** | 0.920 | 0.919 |
| CelebA HQ | 0.904 | - | - | 0.851 | 0.884 | 0.895 | 0.825 | 0.874 | **0.918** | 0.912 | 0.914 |
| Kanji | 0.504 | - | - | 0.678 | 0.431 | 0.576 | 0.839 | 0.664 | **0.844** | 0.891 | 0.888 |
| MNIST | 0.521 | - | - | 0.707 | 0.475 | 0.455 | 0.868 | 0.759 | **0.874** | 0.869 | 0.866 |
| | | | | | PSNR | | | | | | |
| Convallaria | 35.45 | - | - | 35.73 | 36.46 | 36.47 | 34.11 | 36.90 | **37.39** | 36.85 | 36.71 |
| Confocal Mice | 37.95 | - | - | 37.56 | 37.96 | 38.17 | 31.62 | 37.82 | **38.28** | 38.19 | 38.39 |
| 2 Photon Mice | 33.81 | - | - | 33.42 | 33.58 | 33.67 | 25.96 | 33.61 | **34.00** | 34.33 | 34.35 |
| Mouse Actin | 33.64 | - | - | 33.39 | 32.55 | 33.86 | 27.24 | 33.99 | **34.12** | 34.60 | 34.20 |
| Mouse Nuclei | 36.20 | - | - | 35.84 | 36.20 | 36.35 | 32.62 | 36.26 | **36.87** | 37.33 | 36.58 |
| Flywing | 23.45 | 24.67 | - | 24.79 | 22.81 | 24.85 | 25.33 | 25.02 | **25.59** | 25.67 | 25.79 |
| BSD68 | 28.56 | 27.96 | 28.61 | 27.70 | 27.95 | 28.46 | 27.20 | 27.42 | **28.82** | 28.86 | 29.07 |
| Set12 | 29.94 | 28.60 | 29.51 | 28.92 | 29.35 | 29.61 | 27.81 | 28.24 | **29.95** | 30.04 | 30.36 |
| BioID Faces | 33.91 | - | - | 32.34 | 34.05 | 33.76 | 31.65 | 33.12 | **34.59** | 35.04 | 35.06 |
| CelebA HQ | 33.28 | - | - | 30.80 | 31.82 | 33.01 | 31.45 | 31.41 | **33.54** | 33.39 | 33.57 |
| Kanji | 20.45 | - | - | 19.95 | 20.28 | 19.40 | 20.44 | 19.47 | **20.72** | 20.56 | 20.64 |
| MNIST | 15.82 | - | - | 19.04 | 18.79 | 13.87 | 20.52 | 19.06 | **20.87** | 20.29 | 20.43 |

Table 5: **Quantitative pixel-noise removal with HDN.** For all conducted experiments, we compare results in terms of Structural Similarity (SSIM) and PSNR. The best performing method is indicated by being underlined, best performing unsupervised method is shown in bold. For all but one dataset, our HIERARCHICAL DIVNOISING is the new unsupervised SOTA method, even superseding the competing supervised baselines on 4 of the used datasets.

| Datasets | HDN MMSE | HDN Median Estimate |
|---|---|---|
| Convallaria | 37.39 | 37.28 |
| Confocal Mice | 38.28 | 38.25 |
| 2 Photon Mice | 34.00 | 33.91 |
| Mouse Actin | 34.12 | 34.09 |
| Mouse Nuclei | 36.87 | 36.83 |
| Flywing | 25.59 | 25.57 |
| BSD68 | 28.82 | 28.50 |
| Set12 | 29.95 | 29.85 |
| BioID Faces | 34.59 | 34.55 |
| CelebA HQ | 33.54 | 33.51 |
| Kanji | 20.72 | 20.67 |
| MNIST | 20.87 | 20.78 |

Table 6: **Comparison of HIERARCHICAL DIVNOISING MMSE estimate vs HIERARCHICAL DIVNOISING pixel-wise median estimate.** The numbers represent PSNR values. Both the MMSE estimate and median estimates are computed from 100 HDN denoised samples.

a compromise between all possible denoised solutions and hence show faint overlays of diverse interpretations. Note that the MAP estimate gets rid of such faint overlays by committing to the most likely interpretation. For the sake of comparison, we have also showed our MAP estimate to that obtained with DIVNOISING as reported in Prakash et al. (2021).

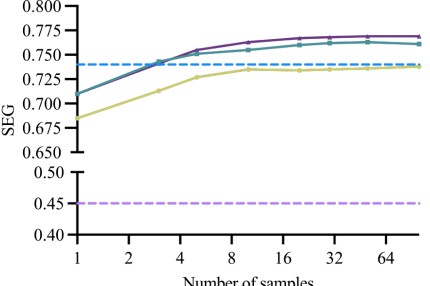 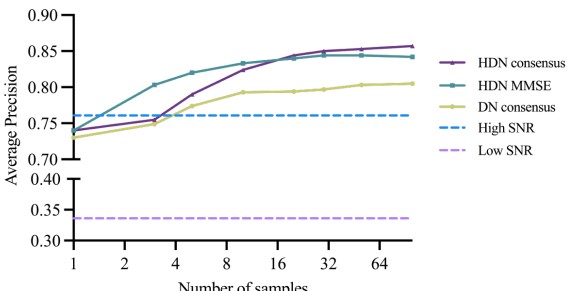

Figure 14: **Application of diverse samples for instance segmentation.** We use the diverse denoised samples from HDN to obtain diverse segmentation results using the procedure presented in Prakash et al. (2021). We then use the diverse segmentation results as input to a consensus algorithm which leverages the diversity present in the samples to obtain a higher quality segmentation estimate than any of the individual diverse segmentation maps. The segmentation performance is compared against available ground truth in terms of Average Precision (Lin et al., 2014) and SEG score (Ulman et al., 2017). Higher scores represent better segmentation. The segmentation performance corresponding to our HDN MMSE estimate obtained by averaging different number of denoised samples improves with increasing number of samples used for obtaining the denoised MMSE estimate. Clearly, the segmentation results obtained with consensus method using our HDN samples outperforms even segmentation on high SNR images using as little as 3 diverse segmentation samples.

## A.8    DOWNSTREAM APPLICATION: INSTANCE SEGMENTATION OF CELLS

Here we present a downstream application of having diverse denoising results. We choose the same instance cell segmentation application for the *DenoiSeg Flywing* (Buchholz et al., 2020) dataset as also presented in Prakash et al. (2021). For a fair comparison to DN (Prakash et al., 2021), we also use the same unsupervised segmentation methodology as described in Prakash et al. (2019a) (locally thresholding images with mean filter of radius 15 followed by skeletonization and connected component analysis) to obtain instance cell segmentations. Thus, we obtain diverse segmentation maps corresponding to diverse denoised solutions.

We then used the label fusion method (*Consensus (Avg.)*) as described in Prakash et al. (2021) which takes the diverse segmentation maps and estimates a higher quality segmentation compared to any individual segmentation. We measure the segmentation performance against available ground truth segmentation in terms of Average Precision (Lin et al., 2014) and SEG score (Ulman et al., 2017) with higher scores representing better segmentation.

For reference, we also show segmentation result when the same segmentation method is applied directly to noisy low SNR and ground truth high SNR images. The segmentation performance on our HDN MMSE estimate obtained by averaging different number of denoised samples improves with increasing number of samples used for obtaining the denoised MMSE estimate. Clearly, the segmentation results obtained with the deployed consensus method using our HDN samples outperform even segmentation on high SNR images. This is true starting with as little as 3 diverse segmentation samples. The performance of the same consensus algorithm with the diverse samples obtained with DIVNOISING (DN) is also shown and is much inferior compared to our results.

## A.9    HARDWARE REQUIREMENTS, TRAINING AND INFERENCE TIMES

***Hardware Requirements.*** The training for all models presented in this work were performed on a single Nvidia Tesla P100 GPU requiring only 6 GB of GPU memory. Hence, our models can be trained relatively cheaply on a commonly available GPU.

***Number of Trainable Parameters and Training Times.*** All our networks have 7.301 million trainable parameters except for the network used for MNIST dataset which has 3.786 million trainable parameters. Thus, our networks have a rather small computational footprint compared to many commonly used generative network architectures such as PixelCNN++ (Salimans et al., 2017), BIVA (Maaløe et al., 2019), NVAE (Vahdat & Kautz, 2020) and VDVAE (Child, 2021) with 53

million, 103 million, 268 million and 115 million trainable parameters, respectively (numbers taken from Child (2021)).

Depending on the dataset, the training time for our models vary between $12 - 24$ hours in total on a single Nvidia Tesla P100 GPU. In comparison, the training of baseline DN (Prakash et al., 2021) takes around $5 - 12$ hours while the supervised CARE (Weigert et al., 2018) method takes around 6 hours for training on the same hardware. Thus, our training times are adding only a small factor compared over DN but we show that we get much improved results (see Fig. 1 and Table 1 in main text). Compared to CARE, our method is unsupervised and hence we do not need high and low quality image pairs for training. In fact, our performance is almost on par with supervised CARE results on most datasets and even outperform CARE on some of them (see Table 1 in main text). An additional advantage of our method over CARE is our ability to generate multiple plausible diverse denoised solutions for any given noisy image which other supervised methods like CARE (or even other unsupervised methods except DN) cannot.

***Prediction/Sampling Times.*** For any dataset, to sample one denoised image with our HDN model, we need at most 0.34 seconds, thus making HDN attractive for practical purposes. The sampling time for one denoised image for all considered datasets is shown in Table 7. Given the samples, the MMSE inference takes 0.000012 seconds only. Hence, computing the MMSE estimate takes almost the same time as computing 100 samples. With DN, the MMSE inference time is around 7 seconds for the largest dataset (compared to 34 seconds with HDN). Although the MMSE inference time with HDN is slightly higher, the performance of HDN is far better compared to DN for all datasets (see Table 1 and Table 3) while still being fast enough for almost all practical purposes. Additionally, HDN allows for interpretable structured noise removal unlike DN.

| Datasets | Time (1 sample) | Time (100 samples) |
|---|---|---|
| Convallaria ($512 \times 512$) | 0.17 sec | 17 sec |
| Mouse Actin ($1024 \times 512$) | 0.34 sec | 34 sec |
| Mouse Nuclei ($512 \times 256$) | 0.11 sec | 11 sec |
| Flywing ($512 \times 512$) | 0.17 sec | 17 sec |
| Confocal Mice ($512 \times 512$) | 0.17 sec | 17 sec |
| 2 Photon Mice ($512 \times 512$) | 0.17 sec | 17 sec |
| BSD68 ($481 \times 321$) | 0.12 sec | 12 sec |
| Set12 ($512 \times 512$ or $256 \times 256$) | 0.13 sec | 13 sec |
| BioID Faces ($384 \times 284$) | 0.10 sec | 10 sec |
| CelebA HQ ($256 \times 256$) | 0.07 sec | 7 sec |
| Kanji ($64 \times 64$) | 0.04 sec | 4 sec |
| MNIST ($28 \times 28$) | 0.02 sec | 2 sec |
| Struct. Convallaria ($512 \times 512$) | 0.17 sec | 17 sec |
| Struct. Mouse ($256 \times 256$) | 0.17 sec | 17 sec |
| Struct. MSC ($512 \times 512$) | 0.08 sec | 8 sec |

Table 7: **Sampling times with HIERARCHICAL DIVNOISING.** Average time (evaluated over all test images) needed to sample 1 denoised image and 100 denoised images from a trained HIERARCHICAL DIVNOISING network. The size of test images is also shown.

### A.10    LIMITATIONS OF HDN FOR STRUCTURED NOISE REMOVAL

As discussed in Sec. 7 in the main text, we expect that our proposed technique of selectively "deactivating" bottom-up layers will be useful for commonly occurring spatial artefacts in microscopy images and thus has wide practical applicability. Microscopy data is diffraction limited and the pixel-resolution of micrographs typically exceeds the optical resolution of true structures. In such situations, the true image signal is blurred by a point spread function (PSF), but imaging artefacts are often not. Hence, the spatial scale of the true signal in microscopy data is not the same as the spatial scale of many structured microscopy noises and HDN captures these structured noises mostly in the bottom two layers. This allows our HDN setup to disentangle the artefacts from true signals in different hierarchical levels, subsequently allowing us to perform selective bottom-up "deactivation" technique (as described in the main text) to remove said artefacts. However, if the spatial scale of

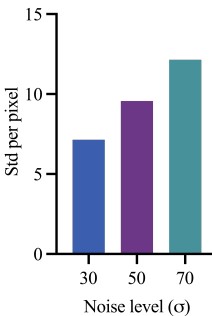

Figure 15: **Measuring sample diversity with increasing noise levels on imput images.** We synthetically corrupt *DenoiSeg Flywing* dataset with different levels of Gaussian noise ($\sigma = 30, 50, 70$) and train different HDN networks for these noise levels. For each of these networks, we sampled 100 HDN denoised solutions corresponding to each noisy image in the test set. For the 100 samples, we computed the standard deviation per pixel and report the average standard deviation over all test images. The higher the noise level, the greater the standard deviation per pixel is. This is expected because more noisy data is more ambiguous and hence HDN accounts for this uncertainty in the increasingly noisy input images by generating more diverse samples.

the image artefacts is at the same scale as that of true signals, our HDN setup will not be able to disentangle the artefacts across scales and hence will not be applicable in such cases.

## A.11 ARCHITECTURAL DIFFERENCES BETWEEN HDN AND DN

As noted in Sec. 4 and illustrated in Fig. 5, our HDN architecture is significantly different from DN. HDN uses several architectural components not present in DN which ultimately contribute to significant performance boosts.

The foremost difference is that DN uses a single layer latent space VAE architecture whereas we employ a multi-layer hierarchical ladder VAE for increased expressivity (see Table 1 and Fig. 2). Unlike DN, our HDN models have hierarchical stochastic latent variables in the *bottom-up* and *top-down* networks with each layer consisting of 5 gated residual blocks. As shown in Table 2, increasing the number of latent variables as well as the number of gated residual blocks per layer increases performance till a certain point, after which the performance more or less saturates. Additionally, unlike DN, we find that employing batch normalization (Ioffe & Szegedy, 2015) significantly enhances performance by about 0.5 dB PSNR on BSD68 (see Table 2). Furthermore, the generative path of our model conditions on all layers above by using skip connections which yield a performance gain of around 0.2 dB PSNR on BSD68 dataset. Finally, to stabilize network training and avoid the common problem of posterior collapse (Bowman et al., 2015) in VAEs, we use the free-bits approach used in Kingma et al. (2016). This is a significant difference from DN which uses KL annealing (Bowman et al., 2015) to avoid this problem. Note that the public code of DN does not use KL-annealing by default, but only switches it on if posterior collapse happens multiple times (default is set to 20 times in the publicly available implementation). However, we always employ the free-bits approach and we never encounter posterior collapse for any dataset tested so far. In fact, we notice that free-bits also brings in performance improvements (around 0.15 dB on BSD68 dataset) in addition to stabilizing the training.

Thus, architecturally there is a significant difference between HDN and DN and all the architectural modifications play an important role in achieving a better overall performance of HDN over DN (see Table 1).

## A.12 HOW DOES NOISE ON IMAGES AFFECT THE SAMPLE DIVERSITY OF HDN?

Noise introduces ambiguity in the data and hence, more noisy images are more ambiguous. Thus, we expect that HDN samples will be more diverse if input images are more noisy. To verify this, we corrupted the *DenoiSeg Flywing* dataset with three levels of Gaussian noise ($\sigma = 30, 50, 70$) and trained three HDN models, respectively. During prediction, for each noisy image in the test set, we sampled 100 denoised images and compared the per pixel standard deviation. We report the average

of the standard deviations over all test images in Appendix Fig. 15. As expected, the higher the noise on the input datasets, the greater the per pixel standard deviation of the 100 denoised samples, indicating higher data uncertainty at higher levels of noise.

