# OpenReview forum: "Interpretable Unsupervised Diversity Denoising and Artefact Removal"
_ICLR.cc/2022/Conference — ICLR 2022 Spotlight_

### Official Review · Reviewer_iaBg · 2021-10-31

**Correctness:** 4
**Technical Novelty And Significance:** 3
**Empirical Novelty And Significance:** 2
**Recommendation:** 6
**Confidence:** 3

**Main Review:**

Strengths:
The paper shows one can improve the performance of DN by introducing the hierarchical representation of the latent variables introduced in VAEs. In addition, the paper shows that the artifact removal can be realized by the analysis of the visualized image patterns corresponding to the latent variables in each layer: By deactivating the layers of latent variables at which the artifacts are encoded in the restoration, one can remove artifacts even though they are spatially structured perturbation of pixel values.

Weaknesses:
Artifacts targeted in the paper are not well characterized. It is not sure if the structured noises would be encoded in the first and/or the second layer(s) of the latent variables independently from the input images. The limitation of the performance of the proposed artifact reduction should be experimentally or theoretically explained more clearly. When the deactivation is not applied, would the HDN always generate similar artifacts independently from the input images or from the network architectures? Is it sure if the artifacts are always enced by the latent variables of layers near the bottom? If the artifacts change depending on the input images as described by Eq.(3) and are encoded by the layers near the top, can the proposed method still reconstruct high quality images even when the layers near the top are deactivated?　It is not clearly demonstrated by the current paper whether the proposed method will show high performance of artifact removal against variety of input images.

By the way, if the reviewer understands correctly, the artifacts handled by the proposed method are not generated in the process of imaging but essentially by the reconstruction from given images. The noises denoted by e in Eq.(1) does not include artifacts (spatially correlated noises) and the noises denoted by n in Eq.(3) include the artifacts. Denoting both e and n as "noise" makes it more difficult to understand the proposed method. Artifacts, structured noises, can be generated even in the process of imaging. It is recommended to denote e and n in different words. It is also recommended to clearly describe that only the artifacts generated in the reconstruction process are handled by the proposed method when describing the related works of structured noise removal.

**Summary Of The Paper:**

The paper proposes an image restoration method that can not only reduce pixelwise noises but also remove artifacts in resultant images. Introducing the idea of hierarchical representation of latent variables analogous to VAEs, the proposed method improves the denoising performance of DivNoising (DN). In addition, the authors propose a method for artifacts removal based on the analysis of the image components represented by the latent variables at each layer. The method requires no pairs of noisy images and the corresponding noise-free signals for training but requires the probability distribution of pixel values conditioned by the corresponding signal strength. The experimental results demonstrate the proposed method outperforms DNs.

**Summary Of The Review:**

This paper improves  Prakash et al.(2021). The reported facts that the hierarchical representation of the latent variables improve the performance and that the artifacts are encoded at the layers near the bottom would have worth sharing. As for the artifact removal, on the other hand, it is not so clear how effective the proposed method is for variety of images and hence the reviewer downgrades the rating.

---

> ### Author Response · Authors · 2021-11-18
> **Response to review (Response 1 of 2)**
>
> We thank the reviewer for their comments which help us to further improve our paper and further clarify a few important things.
>
> We believe that there is some degree of misunderstanding regarding the types of artefacts we show can be removed with our approach. While artefacts beyond microscopy can indeed also be removed with HDN, such experiments are not included in the submitted paper and are beyond the scope of our presented work.
> All structured noises presented in this paper are artefacts that are contained in the raw pixel stream as it is received from the camera or Photon Multiplier Tube. As we point out in Section 6, we believe that the presented approach of ablating the lowest $k$ hierarchy levels will likely always work when the resolution of the image signal is lower than the resolution of correlated pixel noises.
>
> Below we address the questions of the reviewer point-by-point.
>
> 1. ***Characterization of artefacts:*** Structured noises, as they appear in real microscopy data, are not easy to spot for a non microscopy expert. To this end, we have now computed the spatial autocorrelation patterns of the noise (at manually selected background/‘empty’ areas for data without available GT). We have updated Figure 4 in main text to show the spatial autocorrelation pattern of noise and refer to a detailed description of how this was computed in Appendix A.1.2 and Appendix Figure 6.
>
> 2. ***When the deactivation is not applied, would the HDN always generate similar artifacts independently from the input images or from the network architectures?*** We showed experimentally in Appendix A.3 (see Appendix Fig. 11) that if deactivation is not applied, the HDN network (due to its high expressivity) captures and reconstructs the structured noise (which is contained in a given input image) quite well. Hence, the artefacts recovered are not independent of the input image, but instead a part of the input image encoding and hence the final decoded images.
> However, the used network architecture limits what can be encoded in the latent space (e.g. our proposed HDN_{3-6}). Hence, structures as well as noises that can/will be recovered also depend on the used architecture.
>
> 3. ***Are structured noises encoded in the first and/or the second layer(s) of the latent variables independently from the input images?***
> No. This is only the case when the resolution of unwanted structured noises and the resolution of wanted image signals don’t overlap too much. This assumption, as also elaborated on above, is reasonable for many diffraction limited microscopy setups on biological samples, where structured noises are essentially associated to the pixel size of the sensing device (e.g. camera), while the true image signal is smoother with respect to the pixel resolution due to the diffraction limited optics of the microscope.
> In other words, the true image signal is blurred by a (somewhat smooth) point spread function (PSF), but many real-world imaging artefacts are not. In all such cases we strongly believe (and support this by our experiments on three quite diverse real-world datasets), that our approach will lead to excellent unsupervised artefact removal results.
>
> 4. ***Limitation of the performance of the proposed artifact reduction should be explained more clearly:***
> The limitations of HDN are now described more clearly in Appendix A.10 and referred to from the main text in Section 7. It is important to mention that HDN leads to excellent artefact removal results for many real-world microscopy datasets, but that for other structured noises (in other domains) the situation could be different. This goes beyond the scope of this work, and will need additional and quite fundamentally different ideas to be handled correctly (see also Appendix A.10).
>
> 5. ***It is not clearly demonstrated by the current paper whether the proposed method will show high performance of artifact removal against variety of input images:***
> We would like to point out that we have shown the superior performance of our method compared to other existing methods used in practice to remove microscopy image artefacts (on 3 real-world datasets) and pixel noise (on 12 datasets across different domains).
> Please note that the spatial extent of the artefacts on all considered datasets is also quite different (see Fig. 4 in main text, Appendix Fig. 6 and Appendix Sec. 1.1.2), thus rendering the unsupervised artefact removal problem challenging. We present a method that successfully does so.
> Additionally, we want to stress that applications in computational biology is a key area mentioned in the ICLR 2022 call for papers (https://iclr.cc/Conferences/2022/CallForPapers). We strongly believe and demonstrate that our method brings theoretical and practical novelty to this arena.
>
> Other questions answered in follow-up thread below.

---

> > ### Author Response · Authors · 2021-11-18
> > **Response to review (Response 2 of 2)**
> >
> > 6. ***If the artifacts change depending on the input images as described by Eq.(3) and are encoded by the layers near the top, can the proposed method still reconstruct high quality images even when the layers near the top are deactivated?:***  Please note that for microscopy images, it is impossible that the artefacts are encoded by the top layers without being captured in the bottom layers as well. As we reasoned in Sec. 7 of the main text (and above), real-world microscopy data is diffraction limited and the scale of the structured artefacts is different from the scale of signal that we care about. If this does not hold, our method will very likely not be applicable. We have added additional clarifications in Appendix A.10.
> >
> > 7. ***The artifacts handled by the proposed method are not generated in the process of imaging but essentially by the reconstruction from given images?:***  This is not necessarily the case. (See next response below, but also relevant responses above.)
> >
> > 8. ***Clarification on  $e$ and $n$ in Eq. 1 and Eq. 3:*** As we mention right after Eq. 4, the reconstructed signal is affected both by colored noise, and by a structured perturbation introduced by the reconstruction method (term in Eq. 4 depending on $s$). This structured perturbation appears even in the case when the measurements are noiseless, and could be considered a bias on the estimation. Nonetheless, we opt to refer to $n$ as noise for simplicity given that this $n$ models all undesired artifacts on the reconstructed signal (even those that are not stochastic in nature). Please note that all this is discussed in the paragraph following Eq. 4, so we hope that this slight ‘abuse’ of terminology doesn't hamper the comprehension but instead even improves it.

---

> > ### Comment · Reviewer_iaBg · 2021-11-23
> > **Response to the response**
> >
> > Thank you for your response.
> >
> > The authors have addressed all my concerns. From my (a non-microscopy expert's) point of view, the characteristics of the structured noise/artifacts are important for understanding the proposed method and should be described in more detail in the main text (not in the appendix). Anyway, reading the other reviewers' comments, the authors' comments, and the updated paper, I decided to raise my rating.

---

### Official Review · Reviewer_5Ptv · 2021-11-01

**Correctness:** 4
**Technical Novelty And Significance:** 3
**Empirical Novelty And Significance:** 3
**Recommendation:** 8
**Confidence:** 4

**Details Of Ethics Concerns:**

No ethical concerns have been identified while reading the manuscript.

**Main Review:**

For the most part, the authors make claims that are well-supported by either the evidence presented in the paper or by the cited research. The presentation of the work, including the use of language, formulas, and graphics is self-consistent and favors smooth reading. The results discussed in the paper make it clear that the Hierarchical DivNoising is superior to all relevant baselines, sometimes even surpassing the performance of supervised methods.

Although overall the paper is very comprehensive and thorough, there are two main things that are worth considering:
1. Section 3, Non-hierarchical VAE Based Diversity Denoisers, discusses the most notable competitors for HDN method, namely classical VAEs algorithms and DivNoising. Later in Section 4, the authors introduce HDN, describing in detail its architecture, loss function, types of extracted representations. The main differences (from the architectural point of view) between HDN and DN need to be explicitly written out either in this or later sections to help the reader assess the novelty of the discussed approach. Also, the authors did not discuss the difference in the number of trainable parameters that can influence both the training time as well as the performance of the model.
2. Hardware requirements. Multiple times across the paper, the authors highlight the low computational cost of the HDN method compared to other VAE-based approaches. Yet, to the best of my knowledge, no specific data on training times i.e. time for one epoch, inference time on the same size data, total training time have been presented in the text.

A few less critical, yet still relevant comments:
1. Nor all variables used in equations are explained or defined. For example, variable s' from the equation (2), N, xi, and i in equation (6). It is important that every element is properly defined to help the reader follow the main idea.
2. Bottom-up, top-down network definitions are missing. In the first paragraph of Section 4, the authors introduce two new terms that have not been defined anywhere above in the text, namely bottom-up and top-down networks. These terms should be defined and explained before being used, otherwise, they might cause confusion.
3. Some equations would benefit from short intuitive explanations. Equations such as (8) and (12) would be easier to comprehend if they would be supported by short and more intuitive explanations.
4. Structured noise patterns are not clearly visible in Figure 4, which shows the results of removing structured noise. As input images also contain pixel-noise it is hard to observe the structured noise, except for N2V where it is possible to see it. Is it possible to make an example that very clearly illustrates the structured noise in the original images?

**Summary Of The Paper:**

Reviewed work presented a new and very promising approach to unsupervised denoising titled Hierarchical DivNoising. Hierarchical DivNoising is based on earlier published DivNoising architecture. The authors compared the performance of their HDN approach to the closest state-of-the-art unsupervised as well as supervised denoising methods on an impressive set of twelve datasets. In all datasets, the presented method has outperformed other unsupervised alternatives, while sometimes even improving on the performance of supervised counterparts. The authors also show that the layers of the proposed architecture encode interpretable levels of abstraction, which can be used to one's own advantage e.g. to get rid of the structured noise often present on the microscopy images.

**Summary Of The Review:**

The reviewed paper makes claims that appear to be correct, supported by empirical evidence, theoretical insights, and prior literature. The proposed method is based on existing technology but extends in a number of ways, which makes the novelty of the paper significant. The potential impact of this research seems to be substantial, provided that denoising is an important problem in many applications related to computer vision. Overall, this is sound work, which seems to significantly improve the current state-of-the-art.

---

> ### Author Response · Authors · 2021-11-18
> **Response to review**
>
> We thank the reviewer for their valuable comments which helped us to further improve our paper.
>
> Below we address the suggestions by the reviewer point-by-point.
>
> 1. ***Architectural differences between DN and HDN need detailed description:*** Thanks for pointing this out. We have now included a detailed comparison of both architectures in Appendix A.11, and a mention of this in the main body of the revised paper (Section 5).
>
> 2. ***Hardware requirements, training and inference times:*** We have updated the paper, now discussing this in Section 5 and Appendix A.9.
>
> 3. ***Definition of all variables in Eq. 2, Eq. 6 and top-down/bottom-up networks:*** We have now added this in the main text at the appropriate places on pages 3 and 4 (blue paragraphs after the equations and at the first introduction of the top-down/bottom-up networks).
>
> 4. ***Adding short and intuitive explanations of Eq. 8 and Eq. 12 for clarity:*** We have now added this in the main text immediately after the equations. Briefly, Eq. 8 represents the DivNoising loss, which has two components: the reconstruction loss which measures how likely the generated image is given the noise model, while the KL loss incentivizes the encoded distribution to be unit Normal.
> Eq. 12 describes the Hierarchical DivNoising loss which also has two components: the reconstruction loss measuring how likely the generated image is under the given noise model while the KL loss is computed between the conditional prior and approximate posterior at each layer.
>
> 5. ***Can the structured noise in the original images be illustrated?:*** We agree with the reviewer that structured noises, as they appear in real microscopy data, are not easy to spot for a non microscopy expert. To this end, we have now computed the spatial autocorrelation patterns of the noise (at manually selected background/‘empty’ areas for data without available GT). We have updated Figure 4 in main text to show the spatial autocorrelation pattern of noise and refer to a detailed description of how this was computed in Appendix A.1.2 and Appendix Figure 6.
>
> We thank the reviewer again for their feedback and questions. We believe that our updated paper answers the questions and incorporates all suggestions.

---

> > ### Comment · Reviewer_5Ptv · 2021-11-18
> > **Response to the response**
> >
> > Thank you for your response.
> >
> > I have reviewed both the authors' responses as well as an updated version of the manuscript and its appendix. To the best of my knowledge, the authors have addressed all my concerns and suggestions in full, providing enough details in places that had a lack thereof. Hence, I am content with my present recommendation to accept this paper.

---

### Official Review · Reviewer_3vAh · 2021-11-02

**Correctness:** 4
**Technical Novelty And Significance:** 3
**Empirical Novelty And Significance:** 3
**Recommendation:** 8
**Confidence:** 3

**Main Review:**

This paper has the following strong points

- First work to use hierarchical Variational Autoencoder for unsupervised image denoising task
- Yields state-of-the-art results on unsupervised image denoising
- Reconstruction results are more interpretable as compared to competing ones
- Could remove structured noise in some special cases of imaging

The only negative that I found concerning this work is that

Lack of discussion on training and inference efficiency comparisons of the proposed method with others
Recommendation: As I have stated under the strong points, this paper brings in multiple contributions in terms of the first attempt of its kind while yielding state-of-the-art results. Therefore I rate this work as novel enough to get accepted.

I have the following questions for the authors.

There has been no discussion on how much inferior the training and test time of the proposed method as opposed to the competing ones such as DivNoising or even other fully supervised methods. Also, how much time does it take to find the MMSE estimate with 100 samples? I would like authors to have a detailed comparison along these aspects.
DivNoising has reported the MAP estimate to be best performing. What is the plausibility of such a MAP estimate with the proposed HDN framework? I do not find any discussion regarding this possibility.
I would like the authors to address the below mentioned points in the final version of their paper.

Section 6 points out to “line artefacts in Fig. 4”. However for a reader, it is hard to understand what these artefacts are. It would be better to give a description or highlight such artefacts for improved revelation of the same. Also, providing additional visualization of the ground truth image can further improve clarity.

Section 7 says “the situation is more complicated for structured noises in other data domains and is slated for future work.” However, it would have been more informative if authors were to show examples of structured noises in these other domains, and how the proposed method works in these cases. This can help shed more light on the limitations of the proposed method and also for future research directions.

**Summary Of The Paper:**

This work extends the work of DivNoising (Prakash et al., 2021) to build an interpretable approach for unsupervised image restoration. The proposed method, namely Hierarchical DivNosing (HDN), is based on the well-studied concept of hierarchical Variational Autoencoder. This work is the first to use hierarchical Variational Autoencoder for the task of unsupervised image denoising. In doing so, HDN achieves state-of-the-art unsupervised image denoising results on twelve benchmark image datasets. The method is also shown to remove structured artifacts on three real microscopy datasets.

**Summary Of The Review:**

This work advances the state-of-the-art in terms of use-case novelty and performance improvement. Henceforth, the contributions are significant enough for the acceptance

---

> ### Author Response · Authors · 2021-11-18
> **Response to review**
>
> We thank the reviewer for their valuable comments and for appreciating our work in terms of novelty, performance and interpretability of our work compared to other baselines.
>
> The reviewer suggested we report three additional aspects of our work (training and prediction times of our method, assessing the feasibility of MAP estimation, and illustrating the input image artefacts more prominently). We have done so and included these in the updated paper, i.e.
>
> 1. We now measure and compare our training times with DN and the supervised baseline and also report our prediction/MMSE inference times. All results can be found in Appendix A.9, and the prediction times of our method is given in Appendix Table 7. For the slowest (because largest) dataset our method takes 34 seconds to compute 100 samples -- hence, enabling even real-time use in parallel to most common microscopy protocols.
>
> 2. Computing other point estimates, e.g. MAP estimates, are still possible with HDN. We have now included this in Section 5, Appendix A.7 and Appendix Figures 12 and 13.
>
> 3. We agree with the reviewer that structured noises, as they appear in real microscopy data, are not easy to spot for a non microscopy expert. To this end, we have now computed the spatial autocorrelation patterns of the noise (at manually selected background/‘empty’ areas for data without available GT). We have updated Figure 4 in main text to show the spatial autocorrelation pattern of noise and refer to a detailed description of how this was computed in Appendix A.1.2 and Appendix Figure 6.
>
> To address the comment about elaborating the limitations of HDN such that readers can be aware of future directions, we have added clarifying text in Section 7 and Appendix A.10. We would like to acknowledge that HDN leads to good results for many real-world microscopy datasets, but that for other structured noises (in other domains) the situation could be different (see Appendix A.10). This will require additional work and quite fundamentally different ideas to be handled properly.
>
> We believe our updated paper addresses the questions by the reviewer and we want to thank the reviewer again for the valuable feedback on our work.

---

> > ### Comment · Reviewer_3vAh · 2021-11-21
> > **Final comments**
> >
> > Thank you for the response.
> >
> > As authors have addressed all my concerns, I advocate for acceptance of this paper.

---

### Official Review · Reviewer_kqm1 · 2021-11-04

**Correctness:** 4
**Technical Novelty And Significance:** 3
**Empirical Novelty And Significance:** 3
**Recommendation:** 8
**Confidence:** 4

**Main Review:**

Strengths:

The proposed model is a novel unsupervised technique that is competitive or outperforms the state of the art using a multiscale variational autoencoder framework. I believe that this is a very interesting contribution. In addition, the paper provides extensive experiments and ablation analysis that provide interesting insights into the properties of the model. Also, the paper is written clearly.

Weaknesses:

In the abstract and introduction, the authors imply that their method produces meaningful uncertainty quantification because it enables one to sample possible restored images, but at the end of the day they use an average as the final denoised result in order to compare to other methods. The stochastic outputs are not evaluated. It would be very interesting (and would back up the tone of the abstract and introduction) if the authors could at least provide an example where the stochastic inputs are useful in some way. This is not a condition for publication, I think the method is interesting in any case.

**Summary Of The Paper:**

The paper proposes a hierarchical VAE model that produces good denoising results. In addition, the model can be adapted to perform removal of more structured artefacts.

**Summary Of The Review:**

The authors show that a multiscale VAE can produce good unsupervised denoising results, provide meaningful analysis, and show that their technique can be adapted to remove structured artefacts. I think this is a meaningful contribution, and therefore I recommend publication.

---

> ### Author Response · Authors · 2021-11-18
> **Response to review**
>
> We thank the reviewer for their valuable comments and for appreciating our work.
>
> To address the reviewer’s wish for us to demonstrate the utility of diverse solutions, we added three analyses to the updated paper.
>
> 1. We show for a downstream segmentation task, that using diverse samples can lead to much improved results (compared to using point estimates). See new text in Section 5 (blue), Appendix A.8 and Appendix Figure 14.
>
> 2. We show that with increasingly corrupted inputs, the diversity in our samples also increases, which is what one would expect if the diversity indeed correlates with the uncertainty in a given input. We mention this now in Section 5, and show results in Appendix A.12 and Appendix Figure 15.
>
> 3. The fact that we can use diverse samples to compute single point estimates (like the MMSE estimate, or other point estimate that aggregates multiple samples) is a powerful feature. We spell this out in Section 5 and show that MAP solutions and pixel-wise median solutions can also be computed having access to diverse samples (see Appendix A.6, A.7, Appendix Figures 12, 13 and Appendix Table 6).
>
> We do believe that these changes improve the overall quality of our paper and want to thank the reviewer again for the valuable feedback.

---

> > ### Comment · Reviewer_kqm1 · 2021-11-29
> > **Final comment**
> >
> > Thank you for incorporating the additional analyses.

---

### Decision · Program_Chairs · 2022-01-20

**Decision:**

Accept (Spotlight)

**Comment:**

A multi-scale hierarchical variational autoencoder based technique is developed for unsupervised image denoising and artefact removal. The method is shown to achieve state of the art performance on several datasets. Further, the multi-scale latent representation leads to an interpretable visualization of the denoising process.

The reviewers unanimously recommend acceptance.